# TradeMaster: A Holistic Quantitative Trading Platform Empowered by Reinforcement Learning

**Shuo Sun**[*]  **Molei Qin**[*]  **Wentao Zhang**  **Haochong Xia**  **Chuqiao Zong**
**Jie Ying**  **Yonggang Xie**  **Lingxuan Zhao**  **Xinrun Wang**[†]  **Bo An**[†]
Nanyang Technological University, Singapore
{shuo003,haochong001,zong0005,ying0024,xiey0035,zhao0375}@e.ntu.edu.sg
{molei.qin,wt.zhang,xinrun.wang,boan}@ntu.edu.sg
[*]Equal contribution    [†]Corresponding author

## Abstract

The financial markets, which involve over $90 trillion market capitals, attract the attention of innumerable profit-seeking investors globally. Recent explosion of reinforcement learning in financial trading (RLFT) research has shown stellar performance on many quantitative trading tasks. However, it is still challenging to deploy reinforcement learning (RL) methods into real-world financial markets due to the highly composite nature of this domain, which entails design choices and interactions between components that collect financial data, conduct feature engineering, build market environments, make investment decisions, evaluate model behaviors and provides user interfaces. Despite the availability of abundant financial data and advanced RL techniques, a remarkable gap still exists between the potential and realized utilization of RL in financial trading. In particular, orchestrating an RLFT project lifecycle poses challenges in engineering (i.e., hard to build), benchmarking (i.e., hard to compare) and usability (i.e., hard to optimize, maintain and use). To overcome these challenges, we introduce TradeMaster, a holistic open-source RLFT platform that serves as a i) software toolkit, ii) empirical benchmark, and iii) user interface. Our ultimate goal is to provide infrastructures for transparent and reproducible RLFT research and facilitate their real-world deployment with industry impact. TradeMaster will be updated continuously and welcomes contributions from both RL and finance communities.

**Software Repository:** https://github.com/TradeMaster-NTU/TradeMaster

## 1  Introduction

Quantitative trading, which refers to the process of applying mathematical models and computer algorithms to automatically identify trading opportunities [8], has been a popular research topic for decades [3]. With the increasing accessibility of financial data [11] and the development of RL techniques [63, 75, 12, 18], RLFT has made great strides in offering profitable financial trading models [68] in recent years. From an algorithmic perspective, many RLFT algorithms are proposed for core quantitative trading tasks such as portfolio management [77, 76, 80, 30], algorithmic trading [13, 66, 42], and order execution [39, 17, 50], respectively. In addition, a plethora of research has tried to address other key problems alongside the RLFT pipeline, including preprocessing financial data [40, 71], alpha discovery [10, 82], market simulation [41, 60, 9], systematic evaluation [67] and online deployment [11]. However, the vast majority of existing works are formulated, addressed and implemented separately, resulting in a stylized range of methods with limited acknowledgement on the complexities and interdependencies in the RLFT pipeline (as a composite). This leads to an often

37th Conference on Neural Information Processing Systems (NeurIPS 2023) Track on Datasets and Benchmarks.

punishing translational barrier between state-of-the-art RLFT research and investors' usage of RL techniques to earn real financial benefits [29, 11, 68].

**Three Challenges.** To facilitate RLFT research and their industry deployments, we call for a holistic platform for the design, development and evaluation of RLFT applications. Specifically, due to the ever-changing nature of financial markets [49] and high performance variability of RL [25], managing real-world RLFT workflows is non-trivial with the following three challenges:

- First and foremost, the engineering problem is that building sophisticated RLFT pipelines requires significant upfront efforts. For an immature domain (e.g., RLFT), it is common to spend >90% of work on implementation details and <10% of work on core technical questions [58]. As an RLFT researcher, it is seldom the case that enough resources are available to build a high quality code base of the complete RLFT workflow. As a result, a holistic platform that encapsulates all major components of RLFT with comprehensive support of mainstream settings and state-of-the-art algorithms is highly desirable for both academic researchers and industry practitioners.

- Second, the benchmarking problem is that the performance of any component depends on its context. For instance, the profitability of an RLFT algorithm is intimately tied to the features used to train RL agents [10] and the market status during evaluation [51]. However, existing works typically examine the merits of each component individually and arbitrarily configure surrounding settings for only ensuring "all else equal" conditions, which lead to inconsistent reporting of revenues with no general consensus on the relative strength of popular RLFT methods [68]. A reproducible empirical benchmark that honestly reflects interactions of RLFT components is urgently needed.

- Third, the usability problem is incurred by the complexities of RLFT, which requires extensive interdisciplinary knowledge to be properly optimized and maintained in practice. Specifically, many knobs (e.g., hyperparameters) need to be tuned for RLFT methods [33] and regular maintenance (e.g., model retraining with up-to-date market data) is unavoidable due to the significant temporal distribution shift of financial markets [35]. These difficulties are usually insurmountable obstacles for non-expert users (e.g., individual investors) and hinder the wide utilization of RLFT methods. A user-friendly platform where the whole process is highly automated with excellent user interfaces is indispensable for this domain.

**Contributions.** TradeMaster is proposed to tackle the above three challenges in one unified RLFT platform with the following contributions: i) As a software toolkit, it offers open-source implementations of the whole RLFT workflow including 13 real-world financial datasets, high-fidelity RL environments for 6 mainstream quantitative trading tasks, 15 popular RLFT algorithms and dozens of tools for systematic evaluation and visualization. This modular and composable structure enables fast development and decrease the cost of collaboration and code-sharing. ii) As an empirical benchmark, it provides rigorous comparison of state-of-the-art RLFT algorithms to determine their effectiveness across different tasks, financial markets and evaluation metrics. The standardized pipelines of TradeMaster ensure that the comparison is transparent, fair and reproducible. iii) As a user interface, TradeMaster provides multiple options for practical use: an open-source Github repository, a Python package, an online web service with graphical user interface (GUI), multiple Jupyter Notebook tutorials and a cloud version using Colab. In addition, we customize advanced AutoML techniques into TradeMaster to finish many key steps of RLFT (e.g., feature engineering and hyperparameter tuning) in an automatic way. These efforts significantly improve the usability of TradeMaster and make it accessible for users with various background and investment goals.

## 2 Quantitative Trading as a Markov Decision Process

As shown in Figure 2, we formulate quantitative trading tasks as a Markov Decision Process (MDP) following a standard RL scenario, where an agent (investor) interacts with an environment (the financial markets) in discrete time to make actions (investment decision) and get reward (profits). Formally, we define the MDP as a 6-tuple: $(\mathcal{S}, \mathcal{A}, P, R, \gamma, H)$. Specifically, $\mathcal{S}$ is a finite set of states. $\mathcal{A}$ is a finite set of actions. $P : \mathcal{S} \times \mathcal{A} \times \mathcal{S} \longrightarrow [0, 1]$ is s state transition function, which consists of a set of conditional transition probabilities between states.

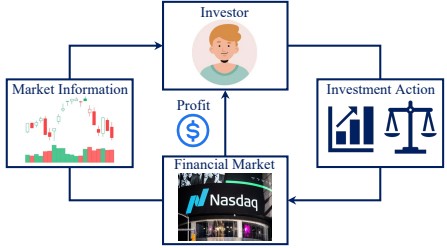

Figure 2: MDP formulation in RLFT

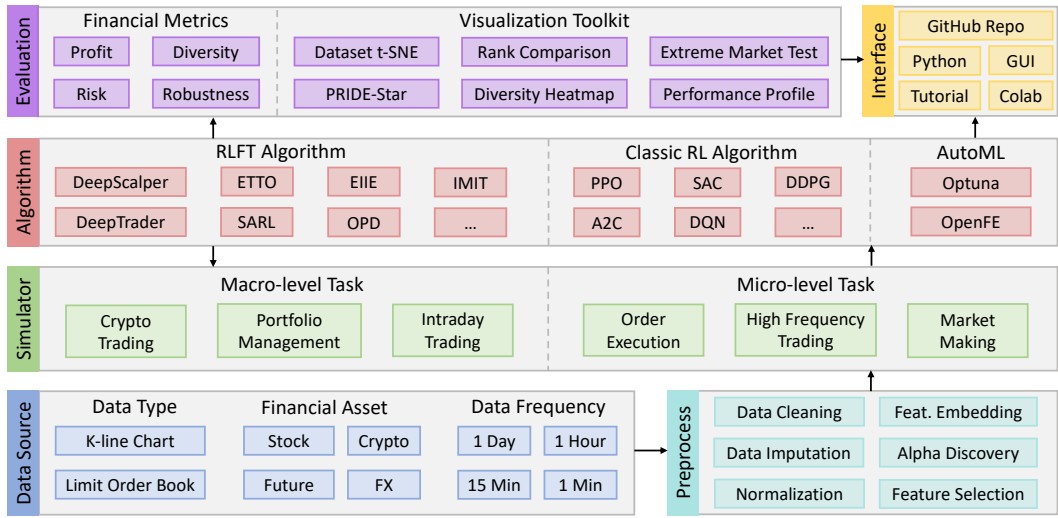

Figure 1: Overview of the TradeMaster platform, which includes six key components for RLFT.

$R : \mathcal{S} \times \mathcal{A} \longrightarrow \mathbb{R}$ is the reward function. $\gamma \in [0, 1)$ is the discount factor and $H$ is a time horizon indicating the length of the trading period. A (stationary) policy $\pi_\theta : \mathcal{S} \times \mathcal{A} \longrightarrow [0, 1]$, parameterized by $\theta$, assigns each state $s \in \mathcal{S}$ a distribution over actions, where $a \in \mathcal{A}$ has probability $\pi(a|s)$. During training, one episode corresponds to making investment decisions at each time step through one whole trading period. The objective of the agent is to learn an optimal policy (investment strategy) that maximizes the expected sum of discounted reward (overall profit): $\pi_{\theta^*} = argmax_{\pi_\theta} \mathbb{E}_{\pi_\theta} \left[ \sum_{i=0}^{T} \gamma^i r_{t+i} \mid s_t = s \right]$. We further describe details on the definition of $\mathcal{S}$, $\mathcal{A}$, $R$ and $P$ in existing works as follows:

**State Space** ($\mathcal{S}$): A state represents RL agents' perception on market status. Specifically, price information (e.g., open, high, low and close price), trading volume, technical indicators and snapshot of limit order book are used in existing works for different trading tasks [76, 66, 17, 78]. In addition, some works feed alternative data, i.e., financial news [56], prediction of future trend [80], experts' investment behaviors [14] and overall market status [77], into states to further improve the performance of RL agents. Furthermore, trader's personal information (e.g., holding position and remaining cash) is also applied as private state in [48, 66, 50] to help investment decision making.

**Action Space** ($\mathcal{A}$): At each time step, the action space represents all possible actions of RL agents' investment decisions that vary under different trading tasks [68]. For single asset trading, the actions are the number of shares to buy/sell or long/short the asset as used in [13, 42]. For portfolio management, the actions represent the proportions of capitals allocated into each financial asset as in [76, 77]. For order execution and market making, the actions correspond to a limit order that shows traders' desired price and quantity as in [17, 39].

**Reward Function** ($R$): Previous works [76, 42, 31] commonly apply the change of capitals (how much money earned/losed) as the reward function, which is natural and easy to implement. Many practical constraints such as commission fee and slippage are further included to be more realistic. In addition, there are also several alternative options designed for specific settings including Sharpe ratio in [77], the hindsight reward function in [66] and the binary sparse reward function in [39].

**Transition Function** ($P$) is defined by the financial market progression following the investment actions. The progression is defined by building realistic data-driven simulated environment based on real-world financial data following the paradigm in [41, 53].

**Examples of the MDP in RLFT.** Considering a simple trading scenario with only one stock, we obtain $p_t^c$ and $p_{t+1}^c$, which denote the close price of the stock at time $t$ and $t + 1$, from historical data. The action at time $t$ is to buy $k$ shares of the stock. Then, the reward $r_t$ at time $t$ is the account profit defined as $k * (p_{t+1}^c - p_t^c)$. For state, we use historical market data to calculate technical indicators as external state and investors' private information such as remaining cash and current position is

applied as internal state. Similar procedures to build RL environments with historical market data have been applied in many RLFT work [40, 77, 80].

**Further Clarification.** We would like to highlight that the above descriptions on the MDP formulation is a general version. As a holistic platform, TradeMaster covers a wide range of trading scenarios, which is impossible to offer detailed MDP formulation for each scenario here due to page limit. For readers unfamiliar with RLFT, Appendix A and this survey [66] offers a detailed introduction on related terminologies and detailed MDP formulation of each RLFT task.

## 3 TradeMaster as a Software Toolkit

To provide a high-quality implementation of the complete RLFT workflow, we first introduce our 3 design principles [26, 29] based on our experience in prototyping and developing RLFT projects. Then, we discuss details on the six component modules of TradeMaster as shown in Figure 1.

---

**TradeMaster Design Principles**

⋆ **Pipeline First, Algorithms Second:** Due to the strict "separation of concerns" produced by the high-level API functions of each component, the goal of TradeMaster is to first enable the proper pipeline functionality, while the intricacies and configurations of individual algorithm choices on the back burner. ⋆ **Be Minimal and Unintrusive:** While workflow development needs to be unified and systematic, we believe learning to use the platform needs to be easy and intuitive. Concretely, this manifests in the clear code structure and user-friendly interfaces of TradeMaster that enables easy usage and adoption for different users. ⋆ **Encourage Extension:** Since RLFT is an emerging research direction with many novel methods proposed rapidly, the platform components should be extensible for the incorporation of new methods. Concretely, TradeMaster encapsulates the design of functions within each component module with support of easy extension.

---

**Data Sources.** We include 13 long-term real-world financial datasets that are highly diverse in terms of financial assets (e.g., stock, FX, and Crypto), financial markets (e.g., US and China), data granularity (e.g., day-level and minute-level) and dataset sizes (e.g., small, medium and large-scale). We summarize statistics of these datasets in Table 1. We host all raw data and corresponding datasheets [23] accessible for users to download on both Google Drive and Baidu Cloud. We further offer scripts for calling API data providers, i.e., Yahoo Finance and AKShare, for users who desire to build personalized datasets. We also plan to further expand our collection of datasets in future updates. See Appendix D for more details on dataset description, collection, maintenance and usage.

Table 1: Dataset statistics of market, frequency, asset amounts, size, chronological period and source

| Dataset | Market | Freq | # of Assets | Size | From | To | Source |
|---------|--------|------|-------------|------|------|-----|--------|
| DJ30 | US Stock | 1 day | 29 | 72994 | 12/01/03 | 21/12/31 | Yahoo |
| SP500 | US Stock | 1 day | 363 | 2009568 | 00/01/01 | 22/01/01 | Yahoo |
| Russell | US Stock | 1 day | 691 | 2391650 | 07/09/26 | 22/06/29 | Yahoo |
| KDD17 | US Stock | 1 day | 41 | 149407 | 07/09/26 | 22/06/29 | Yahoo |
| ACL18 | US Stock | 1 day | 74 | 264954 | 07/09/26 | 22/06/29 | Yahoo |
| SSE50 | China Stock | 1 hour | 26 | 134680 | 16/06/01 | 20/08/30 | Yahoo |
| HSTech | HK Stock | 1day | 30 | 60120 | 88/12/30 | 23/03/27 | AKShare |
| HSI | HK Stock | 1 day | 72 | 206866 | 07/09/27 | 22/06/29 | AKShare |
| Future | Future | 5 min | 20 | 20370 | 23/03/07 | 23/03/28 | AKShare |
| FX | FX | 1 day | 22 | 110330 | 00/01/01 | 19/12/31 | Kaggle |
| USDCNY | FX | 1 day | 1 | 5014 | 00/01/01 | 19/12/31 | Kaggle |
| Crypto | Crypto | 1 day | 1 | 2991 | 13/04/29 | 21/07/06 | Kaggle |
| BTC | Crypto | 1 min | 1 | 17113 | 21/04/07 | 21/04/19 | Binance |

**Preprocessing.** We follow the DataOps paradigm [16, 4, 41] in this component to implement an efficient pipeline for financial data engineering. During data cleaning, we offer scripts to remove duplicated data and merge data from different sources into one unified format. In addition, we apply a conditional score-based diffusion model [71] to conduct data imputation for missing values. For technical indicators, TradeMaster supports both a version of 13 features [81, 69] and Alpha 158 [79], which are widely used for research papers [19, 67] and industry scenarios, respectively. Multiple data

normalization methods are supported, where z-score serves as the default choice. In addition, users can choose to pick a threshold that filters out valuable features with sufficiently large information coefficient on the return for feature selection. After preprocessing, the raw data is converted into high quality ready-to-use data that can be directly used to build realistic data-driven RL environments.

**Environments.** We build data-driven market environments with real-world financial data following the de facto standard of OpenAI gym [6, 41] and the paradigm in [41, 53]. Each environment has three functions: `reset()` function that resets the environment back to the initial state $s_0$; `step()` function that takes an action $a_t$ of the agent and updates state from $s_t$ to $s_{t+1}$; `reward()` function computes the reward value generated by action $a_t$. To make the environments more realistic, we incorporate many practical constraints (e.g.,, transaction costs, slippage and non-negative balance) for different scenarios. There are also options for leveraging, short and loss stopping.

Specifically, the current version supports 6 mainstream quantitative trading tasks [68]: i) For Crypto trading, investors continuously buy/sell cryptocurrencies (e.g., Bitcoin) to make profits. ii) For portfolio management, investors hold a pool of different financial assets and reallocate the proportion of capitals invested in each asset periodically to pursue stable long-term profits. iii) For intraday trading, investors finish the buy/sell or long/short actions within the same trading day to capture the fleeting intraday opportunities and all positions are sold out on market price at the end of the day to avoid overnight risk. iv) Order execution is a micro-level trading task that finish the execution goal within a fixed time horizon (e.g., selling 10 thousand shares of Apple stock in 20 minutes) and try to minimize the costs at the same time. v) Market making focuses on providing liquidity of the market by trading on both buy/sell sides at the same time. vi) High frequency trading utilizes micro-level information and trades in a high-frequency to capture the tiny local price fluctuation.

**Algorithms.** Although there has been extensive works on RL algorithms for different trading scenarios [66, 77, 17], only a tiny fraction of authors choose to make their code publicly available due to the intense competitions and zero-sum game essence in financial markets [72]. As far as we know, TradeMaster is the first open-source platform that offers standardized implementations of a wide range of state-of-the-art RLFT algorithms. For algorithms with official source code, we convert the code into TradeMaster's code structures following the same networks, hyperparameters and other details. This condition fits for SARL [80], DeepTrader [77] and OPD [17]. For other algorithms without publicly available implementations, we reimplement them and try our best to maintain consistency based on the original papers. Brief descriptions of RLFT algorithms implemented in TradeMaster are as follows:

- SARL [80] proposes a state-augmented RL framework, which leverages the price movement prediction as additional states, based on deterministic policy gradient [62] methods.

- DeepTrader (DT) [77] is a policy gradient based methods, which leverages both maximum drawdown and return rate as reward functions to balance between risk and profit.

- EIIE [31] is considered as the first deep RLFT method with an ensemble of identical independent evaluators topology, a portfolio vector memory, and an online stochastic learning scheme.

- Investor-Imitator (IMIT) [14] is an RL approach that imitates behaviors of different types of investors using investor-specific reward functions with a set of logic descriptions.

- DeepScalper (DS) [66] is a risk-aware RL framework for intraday trading, which contains both micro-level and macro-level market embedding, a hindsight reward function to capture long-term trend, and a volatility prediction auxiliary task to model risk.

- ETEO [39] is designed based on the classic proximal policy optimization (PPO) algorithm for order execution with a sparse reward function and a market encoder using LSTM.

- OPD [17] is developed based on ETEO [39] and further adds a policy distillation mechanism to learn from teacher trained based on perfect market information.

Besides advanced RLFT algorithms, TradeMaster also includes 9 efficient customized implementations of classic RL algorithms based on the widely used RLib library [37] including PPO [57], A2C [46], Rainbow [27], SAC [24], DDPG [38], DQN [45], DDQN [74], PG [70], TD3 [21] for various quantitative trading tasks.

**Evaluations.** To provide a systematic evaluation, TradeMaster offers a plethora of evaluation measures and visualization tools based on PRUDEX-Compass [67] to assess RLFT algorithms from

5 axes: profitability, risk-control, universality, diversity and reliability. Specifically, we implement a range of financial metrics including profit metrics (e.g., return rate), risk-adjusted profit metrics (e.g., Sharpe ratio [59]) and risk metrics (e.g., volatility and maximum drawdown). Besides evaluating on regular point-wise financial metrics, we further provide many robust measures and evaluation tools to pursue a comprehensive evaluation. For instance, the performance profile [15] and rank distribution [1] plots are included as two unbiased and robust measures towards reliable RLFT methods. We apply t-SNE [73] and heatmap to show data-level and decision-level diversity in different financial markets. We also provide tools to evaluate RLFT methods under extreme market conditions with black swan events. See Appendix E for more details.

**Interfaces.** TradeMaster provides multiple options for practical use. Specifically, we host an open-source Github repository, build a Python package, serve an online web service with graphical user interface (GUI) [1], offer dozens of hands-on Jupyter Notebook tutorials for quick start and develope a cloud version using Colab to improve accessibility. More details on the usage of different user interfaces are available in Section 5.

## 4 TradeMaster as an Empirical Benchmark

Evaluating any algorithm depends on its context. For instance, how well an RLFT algorithm ultimately performs depends on the choices of many factors such as datasets, preprocessing and evaluation measures. While current RLFT works typically seek to isolate individual gains through "all-else-equal" configurations in experiments, there is often limited overlap in pipeline configurations across different studies. To promote transparency and comparability, TradeMaster aims to serve as a structured evaluation framework by providing a comprehensive RLFT empirical standard. Next, we describe how we benchmark RLFT algorithms using TradeMaster with an demonstrative example of portfolio management task on US stock markets. We first introduce experimental setup (Section 4.1) and analyze the performance of RLFT algorithms (Section 4.2). To offer a more comprehensive analysis, we further report the results of an unbiased and robust measure called rank distribution plots (Section 4.3) and performance on extreme market conditions with black swan events (Section 4.4).

### 4.1 Experimental Setup

We conduct reproducible experiments of the classic portfolio management task on a stock pool of Dow Jones 30 index using the datasets in Table 1 to benchmark results of state-of-the-art RLFT algorithms. Due to limited space, we briefly introduce the experimental setup in the following table with red outlines. More details are available in Appendix F and the TradeMaster Github repository.

Table 2: Performance comparison (mean of 5 individual runs) on the US stock market of 8 RLFT algorithms in terms of 8 financial metrics. Pink and green indicate best and second best results.

|      | TR(%)↑ | SR↑ | CR↑ | SoR↑ | MDD(%)↓ | VOL(%)↓ | ENT↑ | ENB↑ |
|------|--------|------|-------|-------|---------|---------|-------|-------|
| A2C  | 51.92  | 0.750 | 1.510 | 1.468 | 32.44 | 1.373 | 2.161 | 1.373 |
| DDPG | 56.95  | 0.800 | 1.614 | 1.562 | 32.37 | 1.372 | 2.667 | 1.265 |
| EIIE | 49.66  | 0.756 | 1.526 | 1.468 | 30.86 | 1.358 | 3.368 | 1.110 |
| IMIT | -4.95  | 0.102 | 0.240 | 0.234 | 50.52 | 2.078 | 1.460 | 2.149 |
| SARL | 52.13  | 0.752 | 1.544 | 1.490 | 32.23 | 1.420 | 2.758 | 1.192 |
| TD3  | 57.15  | 0.804 | 1.616 | 1.564 | 32.36 | 1.373 | 2.608 | 1.214 |
| PG   | 51.17  | 0.742 | 1.492 | 1.452 | 32.49 | 1.373 | 2.146 | 1.385 |
| PPO  | 50.99  | 0.742 | 1.490 | 1.450 | 32.49 | 1.372 | 2.136 | 1.393 |

### 4.2 Performance Comparison

We report the overall performance of 8 methods in Table 2. TD3 achieves the best profitability with highest values in TR, SR, CR and SoR, while DDPG is also a good option with slightly lower performance. For RL algorithms designed for trading (e.g., EIIE, IMIT and SARL), they fail to outperform properly tuned classic RL algorithms such as TD3 in terms of profitability. For risk-control, EIIE shows best performance with the lowest MDD and VOL. For diversity measures, EIIE

---

[1]http://trademaster.ai

and IMIT perform the best for ENT and ENB, respectively. The overall performance of IMIT is much worse comparing to other methods. One possible reason is the lack of high-quality data of professional investmentors' behaviors [14]. Generally speaking, no existing algorithm achieves dominating results on all financial metrics and there are still vast opportunities to improve in this filed. In addition, we observe that classic RL algorithms (e.g., DDPG and TD3) still outperform RLFT trading algorithms with proper hyperparameter tuning in terms of profit-related metrics (e.g., TR, SR, CR and SoR). At the same time, RLFT algorithms (e.g., EIIE, IMIT and SARL) achieve portfolios with better risk-control and diversity, which demonstrate the effectiveness of the risk-related components. We hope the reproducible benchmark results from TradeMaster can help researchers easily notice the relative strength of existing algorithms and inspire the design of new algorithms.

---

**Experimental Setup of Portfolio Management on US Stock Markets**

**Dataset**: We collect 10-year (2012-2021) historical prices of 29 influential stocks in the Dow Jones 30 index with top unit price as a strong assessment of the US stock market's overall tendency from Yahoo Finance.

**Features**: We generate 11 widely used temporal features based on historical price to describe the stock markets following [81, 67, 69]. Detailed formulas are available in Appendix F.

**Preprocessing**: We conduct z-score normalization of each feature based on the mean and standard deviation of train, validation and test set, respectively. Input missing values are imputed using CSDI [71].

**Training Setup**: We perform all experiments on one RTX 3090 GPU. Each algorithm is trained for 10 epochs for each random seed. We report the average performance of 5 random seeds. It takes about 1 hour to train, validate and test on the US stock market.

**Algorithms**: We compare the performance of 8 state-of-the-art algorithms including 5 classic RL algorithms (A2C [46], DDPG [38], TD3 [21], PG [70] and PPO [57]) and 3 RL algorithms designed for financial trading (EIIE [31], IMIT [14] and SARL [80]) .

**Financial Metrics:** We apply 8 financial metric including 1 profit metric: total return (TR), 3 risk-adjusted profit metrics: Sharpe ratio (SR) [59], Calmar ratio (CR) and Sortino ratio (SoR), 2 risk metrics: volatility (Vol) [61] and maximum drawdown (MDD) [43] and 2 diversity metrics: entropy (ENT) [32] and effect number of bets (ENB) [34]. The mathematical definitions of these metrics are in Appendix E.

**Hyperparameters**: We first select the values of hyperparamters following two conditions: i) if there are authors' official or open-source RLFT library implementations, we apply the same hyperparameters for a fair comparison since they are tuned in financial domain. This condition applies for A2C, DDPG, TD3, PG, PPO and SARL. ii) if there are no publicly available implementations, we reimplement the algorithms and try our best to maintain consistency based on the original papers. This applies for EIIE and IMIT. Later on, we apply grid search on several key RL hyperparameters based on the TradeMaster codebase to further improve performance. Specifically, we try batch size in list [256, 512, 1024], hidden size in range [64, 128, 256] and learning rate in $[3e^{-4}, 5e^{-4}, 7e^{-4}, 9e^{-4}]$ for both actor and critic. Adam is used as the optimizer. More details on the hyperparamters are available in the TradeMaster GitHub repository.

**Data Split**: We follow the rolling data split paradigm in [67] (see figure on the left). Phase three uses the last year for test, penultimate year for validation and the remaining of the dataset for training. For phase one and two, their validation/test sets roll back one and two years.

**Demonstration of Data Split**:

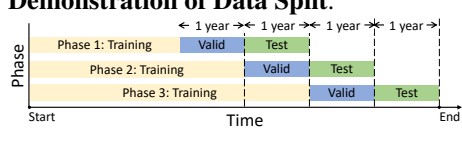

---

## 4.3   Rank Distribution to Demonstrate the Rank of RLFT methods

In Figure 3, we plot the rank distribution plot [1] of 8 RLFT methods in terms of TR, SR, VOL and Entropy across 3 test periods with results of 5 random seeds in each period. It is more robust to outliers comparing to the widely-used mean performance, where the $i$-th column in the rank distribution plot shows the probability that a given method is assigned rank $i$ in the corresponding

metrics. For x-axis, rank 1 and 8 indicate the best and worst performance. For y-axis, the bar length of a given method on a given metric with rank $i$ corresponds to the % fraction of rank $i$ it achieves across the 3 test periods and 5 random seeds ($3 \times 5 = 15$ in total). For TR and SR, TD3 slightly outperforms DDPG with 24% and 27% probability to achieve the best performance. For Vol, EIIE gets the overall best performance while TD3 goes through higher volatility. For ENT, EIIE significantly outperforms other methods with 100% probability for rank 1, which demonstrates its ability to construct portfolios with enough diversity.

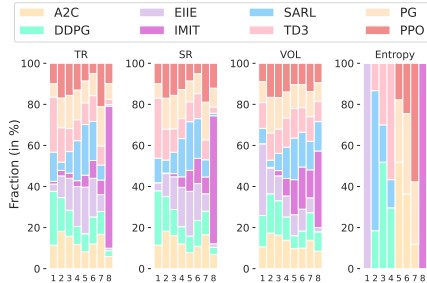

Figure 3: Rank distribution in terms of 4 financial metrics on the US stock market.

## 4.4 The Impact of Extreme Market Conditions

To further provide an evaluation of risk-control and reliability, we pick one extreme market period with black swan events during testing. In this case, the period is September 1-30 in 2021, when the US stock market is violate and goes through the largest decrease after COVID-19 due to the concern on interest rate increase and the congressional shutdown. In Figure 4, we plot the bar chart of TR and SR during the period of extreme market conditions. The red line indicates the market average. Classic RL methods (e.g., A2C, DDPG, TD3, PG, PPO) achieves similar performance comparing to the market average as they have the tendency to keep conservative during extreme market conditions. In contrast, radical methods such as EIIE and SARL are more suitable options with better performance [44]. This analysis on extreme market conditions can shed light on the design of RLFT methods, which is in line with economists' efforts on understanding the financial markets. For instance, researchers may incorporate volatility-aware auxiliary task [66] into the design of RLFT algorithms to make them be aware of extreme market conditions in advance and behave as a profit-seeking agents during extreme market conditions.

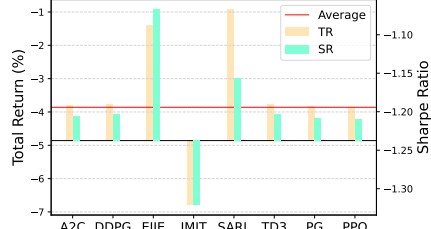

Figure 4: Performance of RLFT methods during extreme market conditions.

---

**Demonstrative Code of TradeMaster Platform**

```
# Load Configuration File
args = parse_args()
cfg = Config.fromfile(args.config)

# Build Environments for Train/Valid/Test
dataset = build_dataset(cfg)
train_env = build_env(cfg, dataset,
   default_args=dict(…, task="train"))
valid_env = build_env(cfg, dataset,
   default_args=dict(…, task="valid"))
test_env = build_env(cfg, dataset,
   default_args=dict(…, task="test"))

# Setup Network and Optimizer
net = build_net(cfg.act)
optimizer = build_optimizer(cfg,
   default_args=dict(…))
```

```
# Setup Loss and Transition Function
criterion = build_loss(cfg)
transition = build_transition(cfg)

# Build Reinforcement Learning Agent
agent = build_agent(cfg,default_args=dict(…))

# Build Trainer Based on Environments
trainer = build_trainer(cfg,default_args=dict(
   train_env=train_env, valid_env=valid_env,
   test_env=test_env, agent=agent))

# The Procedure of Training and Validation
trainer.train_and_valid()

# The Procedure of Testing
trainer.test()
```

## 5 TradeMaster as a User Interface

In this section, we introduce our efforts to make TradeMaster a great user interface. First, we show the clean and extendable code structure of TradeMaster with a demonstrative example. Second, we describe the multiple practical usage options for users from different background in TradeMaster. Third, we discuss on the AutoML component to further improve usability.

**Demonstrative Code.** The code snippet above shows the code of the whole pipeline in TradeMaster including 8 main steps (green code comments), which is simple, clean and easy to extend. We first load

the configuration file and setup the environment, network, optimizer, loss and transition function based on it. Then, we build RL agents and trainer, respectively. Later on, we call `train_and_valid()` function for the training and validation of RL agents. Finally, we call `test()` function to test the performance of RL agents on various financial metrics.

**Multiple Options for Practical Use.** We provide different ways to use TradeMaster for users with different background and investment goals. First, we host a Github repository for TradeMaster including source code, update log and supplementary materials. Second, we provide a Colab version of TradeMaster for users who prefer using cloud computation resources that can run directly. Third, we offer an online web GUI interface. Users can simply pick the settings they are interested in. By clicking on the button, they can train and evaluate RLFT algorithms in a few minutes. This serve as an accessible interface for users with no need on any expert knowledge. Fourth, we provide dozens of Jupyter Notebook tutorials for educational purposes covering key features of TradeMaster.

**Improving Usability with AutoML.** To achieve decent performance, many knobs (e.g., features and hyperparameters) need to be properly tuned for RLFT algorithms [68]. However, this requires in-depth knowledge of RL and coding, which is an insurmountable obstacle for many users without AI background. To make TradeMaster more accessible to these users, we implement an AutoML component [28] to finish the RLFT pipeline in a more automatic way. We aim to provide an interface that enables existing AutoML implementations to be conveniently plugged in. Specifically, we support automatic feature generation and feature selection by customizing the OpenFE [83] library, which firstly generates a set of new features by combining many useful operator (e.g., +, -, min, max) of raw data and then efficiently filters out top-ranked features following a synergistic two-stage evaluation module. In addition, we offer an automatic hyperparameter tuning interface based on Optuna library [2] that enables automatic search for optimal values of learning rate, exploration rate and hidden node amounts. More details are in Appendix G. We note that the current version based on [83, 2] shows the effectiveness of AutoML for the RLFT pipeline and our unified interface design enables easy extension with other advanced AutoML techniques.

## 6 Related Work and Uniqueness of TradeMaster

**Quantitative Trading with Reinforcement Learning.** In the early age, tabular RL is applied for financial trading [47, 48]. Jiang et al. [31] make the first attempt of deep RL in financial markets. FDDR [13] and iRDPG [42] are designed to learn financial trading signals and mimic behaviors of professional traders for algorithmic trading, respectively. For portfolio management, deep RL methods are proposed to account for the impact of market risk [77] and commission fee [76]. A PPO-based framework [39] is proposed for order execution and a policy distillation mechanism is added to bridge the gap between imperfect market states and optimal execution actions [17]. There are also efforts on market making [65, 64], hedging [7], high frequency trading [5] and Crypto trading [52].

Although many works have shown the potential of RL in financial trading, there is still a gap between academic research and real-world deployment. We hope the release of TradeMaster can move one step further toward deployment and facilitate the design of RLFT pipelines.

**Trading Platforms.** There has been many attempts on financial trading platforms. In the early stages, many open-source libraries are proposed for traditional finance approaches [20], event-driven backtesting [55] and portfolio risk analysis [54]. With the advent of AI, recent financial trading platforms mostly focus on data-driven machine learning techniques. OLPS [36] presents a toolbox for online portfolio selection that implements a collection of strategies powered by machine learning. Qlib [79] proposes an AI-oriented quantitative investment platform with a focus on prediction-based supervised learning settings. FinRL [40] makes the first attempt on developing an RL-based quantitative trading platform. FinRL-Meta [41] offers a series of data-driven RL environments for various trading scenarios.

However, due to the existence of alpha decay phenomenon [22], financial practitioners are never keen to share source code of their algorithms and platforms [79] that achieve good performance in real-world settings. TradeMaster is proposed to serve as a first-of-its-kind best-in-class platform with a focus on RL into the family of open-source trading platforms.

**Uniqueness of TradeMaster.** As shown in Table 3, we compare TradeMaster with existing trading platforms [79, 40] to show its uniqueness as follows: i) while existing platforms claims support of various markets, they only offers links to many data providers that users have to write scripts to acquire data flexibly. TradeMaster prepares 13 well-processed datasets and serve them on Google Drive to be downloaded directly. ii) TradeMaster covers a wide range of trading scenarios with realistic simulation under practical constraints. As far as we know, many scenarios (e.g., intraday trading and market making) are not covered by any existing platforms. iii) TradeMaster provides high-quality implementations of 9 classic RL algorithms and 7 RL for trading algorithms, where about half of them is not covered by existing platforms. iv) TradeMaster include a series of evaluation measures and visualization toolkits to provide a systematic evaluation instead of pure financial metrics in existing platforms. v) We build a GUI interface on the official TradeMaster website for users without in-depth coding skills to explore the potential of RLFT. vi) We include an AutoML module for automatic feature selection and hyperparameter tuning to further improve usability.

Table 3: Comparison of TradeMaster and existing trading platforms. # indicates "the number of".

| Platform | # financial markets | # RLFT algorithms | # trading scenarios | # evaluation metrics | # plot toolkits | GUI interface | AutoML |
|---|---|---|---|---|---|---|---|
| Qlib [79] | 2 | 1 | 1 | 4 | - | × | × |
| FinRL [40] | 2 | 8 | 3 | 5 | - | × | × |
| TradeMaster | 13 | 16 | 6 | 10 | 6 | √ | √ |

## 7 Discussion

**Potential Impacts.** We hope that TradeMaster can facilitate the development of the RLFT domain with a broad impact on both academic researchers and financial practitioners. For RLFT researchers, they can rapidly implement and compare their own methods with a focus on real scientific problems instead of engineering details. For RL researchers, TradeMaster introduces many challenging trading scenarios for the test of novel RL algorithms in financial markets. For finance researchers and individual investors, we can have a taste on RL-based trading methods without in-depth knowledge of AI and coding. For professional trading firms, TradeMaster can serve as the code base to enhance their internal trading systems with advanced RL techniques.

**Auxiliary Information.** Due to space limitations, we include some auxiliary yet important information in the Appendix. We introduce the software documents in Appendix C.2, dataset details in Appendix D, official website in Appendix C.1, usage of online GUI in Appendix B, descriptions on evaluation measures in Appendix E, usage of the AutoML component in Appendix G, setup details to reproduce the benchmarking experiments in Appendix F, license and future plan in Appendix H.

**Desiderata.** Machines will never fully replace investors' understanding of the complex economic world, nor researchers' effort on designing novel RL algorithms [29]. TradeMaster enables rapid prototyping, benchmarking and deployment of RLFT procedures, so that investors concentrate on discovering the essential laws of financial markets and researchers can spend more time on core technical challenges. To help grease the wheels, we hope the release of TradeMaster can facilitate the RLFT research and serve as a bridge between finance and AI for broader interdisciplinary impact.

## 8 Acknowledgements

This project is supported by the National Research Foundation, Singapore under its Industry Alignment Fund – Pre-positioning (IAF-PP) Funding Initiative. Any opinions, findings and conclusions or recommendations expressed in this material are those of the author(s) and do not reflect the views of National Research Foundation, Singapore.

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
