# TradeMaster Appendix

**Shuo Sun**   **Molei Qin**   **Wentao Zhang**   **Haochong Xia**   **Chuqiao Zong**
**Jie Ying**   **Yonggang Xie**   **Lingxuan Zhao**   **Xinrun Wang**   **Bo An**
Nanyang Technological University, Singapore

## A  Preliminaries of RLFT

### A.1  Quantitative Trading Preliminaries

In this subsection, we first provide a list of related terminologies of quantitative trading in Table 1. Then, we introduce formal definition of some key terminologies. In addition, readers may refer to [23, 13] for more details.

| Terminologies | Description |
|---|---|
| OHLCV | A abbreviation of open, high, low, close price and volume |
| Limit order book (LOB) | A record of all buy and sell orders for a financial asset |
| Limit order | Orders given to the exchange to buy/sell one asset at a specific price |
| Technical indicator | Mathematical calculation of features to provide insights on financial markets |
| Portfolio | A collection on proportion of capitals invested into different financial assets |
| Pnl | A measure that calculates the net profit or loss generated by a strategy |
| Liquidity | It refers to how easy one asset can be bought or sold in the financial market |
| Black swan event | Extremely rare events that have a significant impact on financial markets |
| DataOps [2] | Principles and rules for the improvement of data quality in data science |
| Temporal distribution shift | The change of statistical properties of data across time |
| Market impact | The impact of investors' actions on the financial markets |
| Signal-to-noise ratio | A ratio of useful signal (high-quality data) to unuseful signal (noise) |
| Market capital | The total value of a publicly traded company |
| Commission fee | The payment made to brokers for executing the trading actions |
| Slippage | The gap between expected price and executed price due to market volatility |
| Market maker | Participants who provide liquidity by trading on both buy/sell sides |
| Foreign exchange | The decentralized global market where currencies are traded |
| Cryptocurrency | A virtual form of currency that utilizes cryptography on blockchain |

Table 1: List of terminologies for quantitative trading

**OHLCV** is a type of bar chart directly obtained from the financial market as shown in Figure 1 (a). OHLCV vector at time $t$ is denoted as $\mathbf{x}_t = (p_t^o, p_t^h, p_t^l, p_t^c, v_t)$, where $p_t^o$ is the open price, $p_t^h$ is the high price, $p_t^l$ is the low price, $p_t^c$ is the close price and $v_t$ is the volume.

**Limit Order** is an order placed to trade a certain number of shares at a specific price. It is defined as a tuple $l = (p_{target}, \pm q_{target})$, where $p_{target}$ represents the submitted target price, $q_{target}$ represents the submitted target quantity, and $\pm$ represents the trading direction (long/short).

**Limit Order Book (LOB)** contains public available aggregate information of limit orders by all market participants as shown in Figure 1 (b). It is widely used as market microstructure [14] in finance to represent the relative strength between the buy and sell side. We denote an $m$ level LOB at time $t$ as $\mathbf{b}_t = (p_t^{b_1}, p_t^{a_1}, q_t^{b_1}, q_t^{a_1}, ..., p_t^{b_m}, p_t^{a_m}, q_t^{b_m}, q_t^{a_m})$, where $p_t^{b_i}$ is the level $i$ bid price, $p_t^{a_i}$ is the level $i$ ask price, $q_t^{b_i}$ and $q_t^{a_i}$ are the corresponding quantities.

37th Conference on Neural Information Processing Systems (NeurIPS 2023) Track on Datasets and Benchmarks.

**Portfolio.** is a vector of weight of each asset that can be represented as:

$$\mathbf{w_t} = [w_t^0, w_t^1, ..., w_t^M] \in R^{M+1} \quad and \quad \sum_{i=0}^{M} w_t^i = 1 \tag{1}$$

where $M+1$ is the number of portfolio's constituents, including cash and $M$ financial assets. $w_t^i$ represents the ratio of the total capitals invested at time $t$ on asset $i$. Specifically, $w_t^0$ represents cash.

**Profit and Loss (Pnl)** indicates the change of total capital, which is widely used as reward for RLFT.

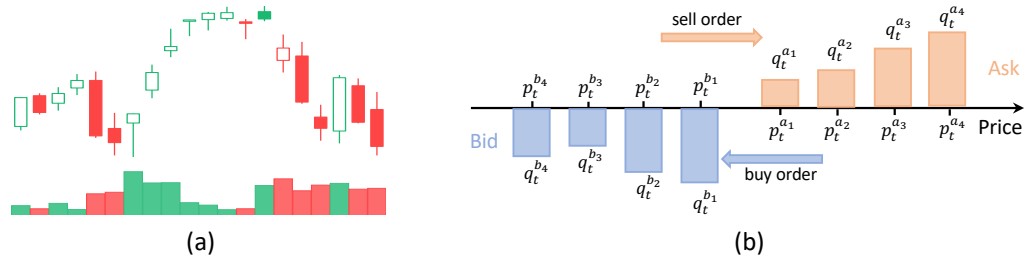

Figure 1: (a) an example of OHLCV candle chart (b) a snapshot of level-4 LOB

## A.2 Reinforcement Learning Preliminaries

For readers interested in reinforcement learning, they can refer to this classic textbook [24] and this survey [10]. We provide a brief description on related concepts in Table 2.

| Terminologies | Description |
|---|---|
| Agent [24] | A reinforcement learning decision maker |
| Environment [24] | A simulation of the financial market that the agent interacts with |
| Gym environment | The standard implementation of RL environments from OpenAI |
| MDP | The mathematical formulation to model decision-making problems |
| Policy | The RL agents' strategies for decision making |
| RLib [11] | A popular open-source library for reinforcement learning |
| DQN [15] | Deep Q-Network that applies neural network to approximate the Q-function |
| DDPG [12] | The deep deterministic policy gradient algorithm |
| A2C [16] | The advantage actor-critic algorithm |
| PPO [19] | The proximal policy optimization algorithm |
| SAC [6] | The soft actor-critic algorithm with entropy regularization |
| DDQN [27] | An enhanced version of DQN to address the overestimation issue |
| PG [25] | The policy gradient algorithms that directly optimize the policy |
| TD3 [4] | The twin delayed deep deterministic policy gradient algorithm |
| AutoML | Methods that automates the procedures of building ML models |
| Hyperparameter tuning | Tuning hyperparameters during training to get better results |

Table 2: List of terminologies for reinforcement learning.

## B  TradeMaster Online GUI Service

TradeMaster provides an online GUI web service hosted on the TradeMaster official website at http://trademaster.ai/. By interacting with the GUI, users can conveniently train and test their own RL agents with no requirements on coding and RL skills.

### B.1  System Overview

The web service is designed and implemented with two parts: ii) the frontend API and ii) the backend server . Here are descriptions on the core components:

**Frontend Part**

- **Experiment configuration** includes the web page GUI, where users can flexibly choose the configurations of experiments. In the current version, we support 3 trading tasks on 3 financial markets with 10 RLFT algorithms.
- **Training part** includes the web page GUI to show the logs of training procedures and visualize validation results.
- **Evaluation part** includes the web page GUI to show the table of performance on test set and visualize test results.

**Backend Part**

- **Start session.** A new session will be created when the backend receives a new training request from the frontend.
- **Get setups.** The backend receives the setups on task, dataset and algorithms from the frontend's request.
- **Build configuration.** A configuration file contains all information is built based on the setup.
- **Start training.** The backend server will train the RL agent based on the configuration file. Then, the performance on validation dataset is evaluated every epoch. Finally, we visualize the net curve based on the best checkpoint and send back the results to frontend.
- **Start evaluation.** We apply the best model on evaluation set to unseen test set. The evaluate performance is sent back to the frontend.
- **Health check.** This part is used to check if the session is running properly. A warning will be generated if the session meets any error.

## B.2 Demonstrative Usage

In this subsection, we show an example of using the TradeMaster GUI online service for training and testing RL agents. As shown in Figure 2, users can pick the task, financial market and RL algorithm by clicking the corresponding buttons. Later on , we click on the "train" button to start the session for training. The backend server will automatically call the functions to train and test the RL algorithms. The final results are shown in Figure 3. We believe the TradeMaster GUI online service is accessible to all types of users and can further benefit the wide distribution of TradeMaster.

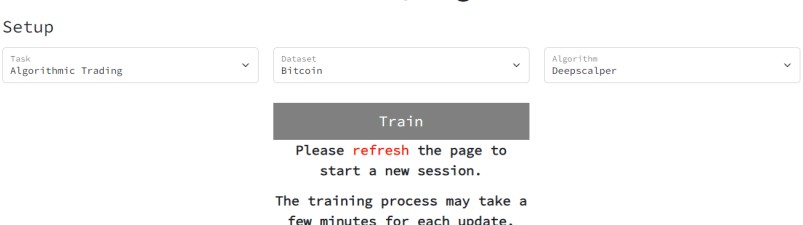

Figure 2: Experiment configuration page of TradeMaster GUI online service

## B.3 Future Improvement

To further improve the online web service, we are currently developing and testing the following features:

- Include pretraining and finetuning paradigm to improve efficiency and generalization ability
- Support more financial markets, trading tasks and parameter options
- Build an evaluation sandbox with various evaluation metrics and result figures
- Support market dynamics modelling and evaluation on user-specific market conditions

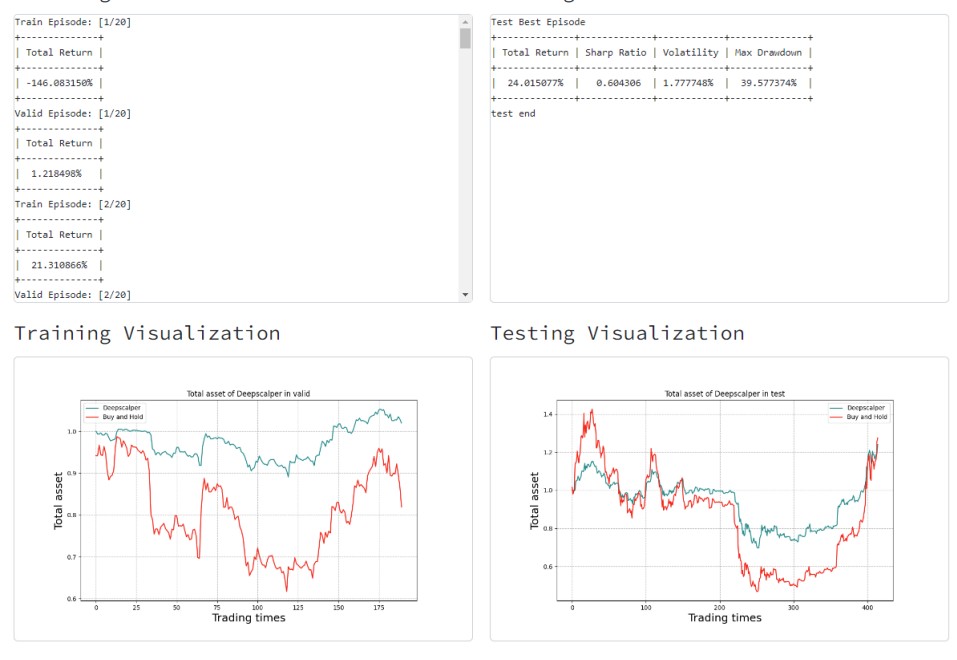

Figure 3: Demonstrative results of training and testing

## C  TradeMaster Ecosystem

### C.1  Official Website

The official TradeMaster website is served at http://trademaster.ai/. There are 4 main web pages: i) the home page contains basic introduction of the project and one button that redirect users to the web service for training RL agents; ii) the service page contains the web service page, where users can flexibly pick preferred settings and train RL agents in a few minutes. iii) The research page includes information of peer reviewed papers we published on RLFT topics; iv) The contact us page contains information of the team and the project email, twitter and zhihu.

### C.2  Software Document

To help users quickly understand the project, we write up software documents of TradeMaster at https://trademaster.readthedocs.io/en/latest/. We first introduce the project overview, installation on multiple operating systems and the model zoo. Then, we discuss on the tutorials and scripts of the core features in TradeMaster. Finally, we include supporting documents such as publication lists and file structures of the project.

### C.3  Useful Tutorials and Scripts

We provide dozens of Jupyter notebook tutorials for educational purposes in TradeMaster at https://github.com/TradeMaster-NTU/TradeMaster/tree/1.0.0/tutorial. A brief introduction of them is as follows:

- **Tutorial 1:** We implement the EIIE algorithm for portfolio management on US stocks.
- **Tutorial 2:** We implement the DeepScalper algorithm for intraday trading tasks.
- **Tutorial 3:** We implement the SARL algorithm for portfolio management on US stocks.
- **Tutorial 4:** We implement the PPO algorithm for portfolio management on US stocks.
- **Tutorial 5:** We implement the ETTO algorithm for order execution.
- **Tutorial 6:** We implement the double DQN algorithm for high frequency trading.

- **Tutorial 7:** We provide a tutorial to show how to automatically tune hyperparameters with the autoML tools in TradeMaster.

- **Tutorial 8:** We provide a tutorial to show the impact on training RL agents with different technical indicators.

- **Tutorial 9:** We provide a tutorial to demonstrate the effectiveness of automatic feature generation and selection.

### C.4 Competitions

We have successfully hosted an RL-based quantitative trading competition called TradeMaster Cup with a total prize of 5000 SGD. More than 40 teams participated in the competition. The competition link is as follows: https://codalab.lisn.upsaclay.fr/competitions/8440?secret_key=51d5952f-d68d-47d9-baef-6032445dea01#learn_the_details

### C.5 Activities on Academic Conference

We have participated and organized many academic activities on top-tier AI conferences with a focus on RL-based quantitative trading based on the TradeMaster project:

- We will host the AAAI 2023 summer symposium on AI for Fintech in Singapore: https://sites.google.com/view/aaai23-ai4fintech

- We are invited to give a talk on TradeMaster in KDD 2023 Finance Day: https://kddfinanceday.github.io/speakers/

- We have provided a one-hour tutorial on RL for trading at AAAI-23 bridge event on AI for financial services: https://sites.google.com/view/aaai-ai-fin/home

- We are invited to provide a tutorial on RL for trading at IJCAI-23: https://ijcai-23.org/tutorials/

## D   Dataset Documents

### D.1   Datasheets for dataset

We further prepare dataset documents using the recommended datasheets [5] to provide details on motivation, composition, collection, preprocessing, uses, distribution and maintenance.

**Motivation**

- **For what purpose was the dataset created?**
  To fulfill the requirements of different TradeMaster users, we aim to build a colorful collection of real-world financial data as the foundation of different trading scenarios and RLFT methods.

- **Who created the dataset?**
  TradeMaster is an open-source project created by the AMI group at Nanyang Technological University. Datasets of TradeMaster are collected and contributed by the authors of this work and will be continuously maintained.

- **Who funded the creation of the dataset?**
  TradeMaster is supported by the National Research Foundation, Singapore under its Industry Alignment Fund – Pre-positioning (IAF-PP) Funding Initiative. Any opinions, findings and conclusions or recommendations expressed of the paper are those of the author(s) and do not reflect the views of National Research Foundation, Singapore.

**Composition**

- **What do the instances that comprise the dataset represent?**
  The instances of TradeMaster datasets include real-world financial data of different financial assets (e.g., stock, FX, and Crypto), financial markets (e.g., US and China), data granularity (e.g., day-level and minute-level).

- **How many instances are there in total?**
  As shown in the size column of Table 3, TradeMaster includes a wide range of financial datasets with varying sizes. The smallest dataset is the Crypto that contains 2991 instances. The largest dataset is Russell that contains 2391650 instances of 691 stocks.

- **Does the dataset contain all possible instances or is it a sample of instances from a larger set?**
  Yes, the datasets are collected from many open-source financial data service providers as concrete examples for direct usage. We also provide scripts and tools to help users acquire data from the original source by flexibly specifying the time period, asset pool, etc.

- **What data does each instance consist of?**
  We summary the dataset statistics of market, frequency, asset amounts, size, chronological period and source in Table 3.

Table 3: Dataset statistics of market, frequency, asset amounts, size, chronological period and source

| Dataset | Market | Freq | # of Assets | Size | From | To | Source |
|---|---|---|---|---|---|---|---|
| DJ30 | US Stock | 1 day | 29 | 72994 | 12/01/03 | 21/12/31 | Yahoo |
| SP500 | US Stock | 1 day | 363 | 2009568 | 00/01/01 | 22/01/01 | Yahoo |
| Russell | US Stock | 1 day | 691 | 2391650 | 07/09/26 | 22/06/29 | Yahoo |
| KDD17 | US Stock | 1 day | 41 | 149407 | 07/09/26 | 22/06/29 | Yahoo |
| ACL18 | US Stock | 1 day | 74 | 264954 | 07/09/26 | 22/06/29 | Yahoo |
| SSE50 | China Stock | 1 hour | 26 | 134680 | 16/06/01 | 20/08/30 | Yahoo |
| HSTech | HK Stock | 1day | 30 | 60120 | 88/12/30 | 23/03/27 | AKShare |
| HSI | HK Stock | 1 day | 72 | 206866 | 07/09/27 | 22/06/29 | AKShare |
| Future | Future | 5 min | 20 | 20370 | 23/03/07 | 23/03/28 | AKShare |
| FX | FX | 1 day | 22 | 110330 | 00/01/01 | 19/12/31 | Kaggle |
| USDCNY | FX | 1 day | 1 | 5014 | 00/01/01 | 19/12/31 | Kaggle |
| Crypto | Crypto | 1 day | 1 | 2991 | 13/04/29 | 21/07/06 | Kaggle |
| BTC | Crypto | 1 min | 1 | 17113 | 21/04/07 | 21/04/19 | Binance |

- **Is there a label or target associated with each instance?**
  No, there is no label or target associated with each instance as our focus is not supervised learning settings.

- **Is any information missing from individual instances?**
  Yes, it is common to have missing values in financial datasets. We provide scripts to preprocess and conduct data imputation with diffusion models [26].

- **Are relationships between individual instances made explicit?**
  Yes, each instance corresponds to the information of one particular asset with timestamp.

- **Are there recommended data splits?**
  For data split, we recommend the similar split procedure in [18, 22] with rolling window for all four datasets. As shown in Figure 4, phase 3 uses the last year for test, penultimate year for validation and the remaining of the dataset for training. For phase one and two, their validation/test sets roll back one and two years, respectively.

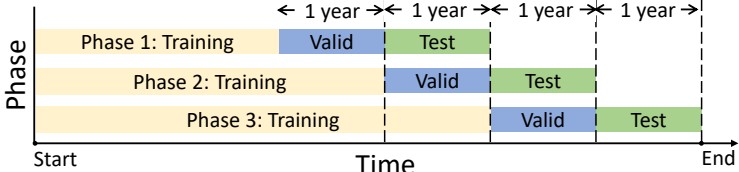

Figure 4: Train/valid/test split procedure with rolling windows

- **Are there any errors, sources of noise, or redundancies in the dataset?**
  There are noise, redundancies and outliers in the raw data collected from different sources. We provide codes and scripts for data cleaning and the preprocess data can be used to build standard RL environment for trading.

- **Is the dataset self-contained, or does it link to or otherwise reply on external resources?**
  Yes, the dataset is self-contained. Users can easily download them from our Google Drive and Baidu Cloud links.

- **Does the dataset contain data that might be considered confidential?**
  No, all TradeMaster data is collected from publicly available sources.

- **Does the dataset contain data that, if viewed directly, might be offensive, insulting, threatening, or might otherwise cause anxiety?**
  No, all data are real-world financial data with numerical values.

**Collection Process**

- **How was the data associated with each instance acquired?**
  The data is acquired from the data sources as shown in the source column of Table 3.

- **If the dataset is a sample from a large set, what was the sampling strategy?**
  We pick the datasets in a dynamic way and try to cover different financial market conditions. Users can also use the scripts we provided to build datasets based on their own preferences.

- **Who was involved in the data collection process and how were they compensated?**
  The TradeMaster team members write scripts for the data collection. All of them is publicly available and free for users.

- **Over what timeframe was the data collected?**
  It varies for different datasets. Most datasets are collected in a daily basis and some datasets are collected by running scripts in real time markets.

- **Were any ethical review process conducted?**
  No, it is not necessary in our case as all data is publicly available financial data.

**Preprocessing/cleaning/labeling**

- **Was any preprocessing/cleaning/labeling of the data done?**
  Yes, we provide codes and scripts for raw data preprocessing and cleaning. More details are available in Section 3.

- **Was the "raw" data saved in addition to the preprocessed/cleaned/labeled data?**
  Yes, all raw data is available at the TradeMaster datasets Google Drive:
  https://drive.google.com/drive/folders/19Tk5ifPz1y8i_pJVwZFxaSueTLjz6qo3

- **Is the software that was used to preprocess/clean/label the data available?**
  Yes, all related codes and scripts are available in the preprocessing component of TradeMaster.

**Uses**

- **Has the dataset been used for any tasks already?**
  Yes, TradeMaster has been used for 6 mainstream quantitative trading tasks.

- **Is there a repository that links to any or all papers or systems that use the dataset?**
  Research papers related to TradeMaster are listed in this reposiotry:
  https://github.com/TradeMaster-NTU/fintech-literature
  The evaluation component of TradeMaster is largely developed based on:
  https://github.com/TradeMaster-NTU/PRUDEX-Compass

- **What tasks could the dataset be used for?**
  Even although we collect the datasets for RLFT, they are also be used for other Fintech settings such as price prediction, risk management and market dynamics modelling.

- **Is there anything about the composition of the dataset or the way it was collected and preprocessed/cleaned/labeled that might impact future uses?**
  We believe that users will be encounter usage limit in the future as all TradeMaster data is collected from publicly available sources.

- **Are there tasks for which the dataset should not be used?**
  We encourage users to use TradeMaster datasets for any task as long as it is legaltimate.

**Distribution**

- **Will the dataset be distributed to third parties outside of the entity (e.g., company, institution, organization) on behalf of which the dataset was created?**
  No, we host the dataset under Apache 2.0 License for academic research.

- **How will the dataset will be distributed?**
  We distribute the dataset through Goolge Driven and Baidu Cloud available at:
  `https://drive.google.com/drive/folders/19Tk5ifPz1y8i_pJVwZFxaSueTLjz6qo3`
  `https://pan.baidu.com/share/init?surl=njghvez53hD5v3WpLgCg0w` (extraction code x24b)
- **When will the dataset be distributed?**
  The first version of TradeMaster is released on March, 2023.
- **Will the dataset be distributed under a copyrightor other intellectual property (IP) license, and/or under applicable terms of use (ToU)?**
  No, TradeMaster is distributed under Apache 2.0 License.
- **Have any third parties imposed IP-based or other restrictions on the data associated with the instances?**
  N/A
- **Do any export controls or other regulatory restrictions apply to the dataset or to individual instances?**
  No

**Maintenance**

- **Who will be supporting/hosting/maintaining the dataset?**
  The TradeMaster team will keep actively supporting/hosting/maintaining the dataset to benefit the rapidly growing RLFT community.
- **How can the owner/curator/manager of the dataset be contacted?**
  We recommend users to contact us by opening issues on GitHub: `https://github.com/TradeMaster-NTU/TradeMaster/issues/new` or sending emails to: `TradeMaster.NTU@gmail.com`
- **Is there an erratum?**
  Users can report issues, confusions and bugs through GitHub. Our team members will respond as soon as possible to help.
- **Will the dataset be updated?**
  Yes, we plan incldue more datasets and improve the whole pipeline based on users' requirements and feedback. Please pay attention to the GitHub repository for lastest update.
- **If the dataset relates to people, are there applicable limits on the retention of the data associated with the instances?**
  N/A
- **Will older versions of the dataset continue to be supported/hosted/maintained?**
  Yes, all versions will be available in the TradeMaster GitHub repository
- **If others want to extend/augment/build on/contribute to the dataset, is there a mechanism for them to do so?**
  We welcome all users to contribute to TradeMaster. One reasonable way is to make a pull request through GitHub.

### D.2 Dataset Descriptions

DJ30 is a dataset containing 10-year (2012-2021) day-level historical prices of 29 component stocks of DJ30 Index, which represents top 30 prominent companies listed on stock exchanges in the US.

S&P500 is a dataset containing 21-year (2000-2021) day-level historical prices of 363 component stocks of S&P500 Index, which tracks the performance of 500 of the largest companies listed on stock exchanges in the US.

Russell is a dataset containing day-level historical prices of 691 component stocks of the Russell 1000 Index. The stock pool contains the highest-ranking 1,000 stocks in the Russell 3000 Index, which seeks to be a benchmark of the entire US stock market.

ACL18 [28] is a widely used public dataset collected using Yahoo Finance[1], which contains day-level historical stock prices of 88 US stocks with top capital size from 9 different industry categories.

---

[1] Yahoo Finance: https://github.com/yahoo-finance

KDD17 [31] is a widely used public dataset collected using Yahoo Finance, which contains 10-year day-level historical stock prices of 50 US stocks selected from 10 different industry categories. In our version, we select a wide time range of 16 years (2007-2022) and 9 stocks are dropped from the pool because of missing values.

SSE50 is a dataset of 4-year (2016-2020) hour-level historical prices of 47 Chinese stocks collected from Yahoo Finance. The stock pool contains component stocks of SSE50 Index, which represent top 50 companies by "float-adjusted" capitalization in Shanghai exchange.

HSI is a datasest collected from AKShare[2] containing historical prices of 72 component stocks of the Hang Seng Index, which tracks the behaviors of the largest companies listed on Hong Kong Stock Exchange.

HSTech is a dataset collected from AKShare containing historical stock prices of 30 largest technology companies listed in Hong Kong that have high business exposure to technology themes.

Future is a dataset collected from AKShare containing 5min-level historical future prices of 20 mainstream countries' currencies in March 2023.

FX is a dataset with 20-year (2000-2019) day-level historical trading price of 22 currencies to USD.

USDCNY is a dataset with 20-year (2000-2019) day-level historical trading price of CNY/USD.

Crypto is a dataset with 9-year (2013-2021) day-level historical price of BTC to US dollar.

BTC is a dataset with minute-level limit order book history of BTC trading in two weeks of April 2023 collected from Binance API[3].

## E   Descriptions on Evaluation Measures in TradeMaster

The evaluation component of TradeMaster is evolved based on PRUDEX-Compass [22] with dozens of evaluation measures and visualization toolkits.

**Profit** measure contains metrics to evaluate FinRL methods' ability to gain market capital. Total return (TR) is the percent change of net value over time horizon $h$. The formal definition is $TR = (n_{t+h} - n_t)/n_t$, where $n_t$ is the corresponding value at time $t$.

**Alpha Decay** indicates the loss in the investment decision making ability of FinRL methods over time due to distribution shift in financial markets. In finance, information coefficient (IC) across time is widely-used to measure alpha decay [17].

**Equity Curve** is a graphical representation of the value changes of trading strategies over time. An equity curve with a consistently positive slope typically indicates that the trading strategies of the account are profitable. In RL settings, we usually plot equity curves with mean and standard deviation of multiple random seeds.

**Risk** includes a class of metrics to assess the risk level of FinRL methods.

- **Volatility (Vol)** is the variance of the return vector $\mathbf{r}$. It is widely used to measure the uncertainty of return rate and reflects the risk level of strategies. The definition is $Vol = \sigma[\mathbf{r}]$

- **Maximum drawdown (MDD)** measures the largest decline from the peak in the whole trading period to show the worst case. The formal definition is $MDD = \max_{\tau \in (0,t)}[\max_{t \in (0,\tau)} \frac{n_t - n_\tau}{n_t}]$

- **Downside deviation (DD)** refers to the standard deviation of trade returns that are negative.

**Risk-adjusted Profit** calculates the potential normalized profit by taking one share of the risk. We define three metrics with different types of risk:

- **Sharpe ratio (SR)** is a risk-adjusted profit measure, which refers to the return per unit of deviation: $SR = \frac{\mathbb{E}[\mathbf{r}]}{\sigma[\mathbf{r}]}$

[2]AKShare: https://akshare.xyz/
[3]Binance: https://www.binance.com/

- **Sortino ratio (SoR)** is a variant of risk-adjusted profit measure, which applies DD as risk measure: $SoR = \frac{\mathbb{E}[\mathbf{r}]}{DD}$

- **Calmar ratio (CR)** is another variant of risk-adjusted profit measure, which applies MDD as risk measure: $CR = \frac{\mathbb{E}[\mathbf{r}]}{MDD}$

**Extreme Market.** It is necessary to evaluate on extreme market conditions with black swan events to show the reliability of FinRL methods .By analyzing the trading behaviors during extreme markets, we can understand their cons and pros and further design better FinRL methods. There are a few potential testbed such as COVID-19 pandemic, financial crisis, government regulation and war.

**Countries.** Financial markets in different countries have different trading patterns, where markets in developed countries is more "effcient" with high proportion of institutional investors and markets in developing countries is more noisy with high personal investors. It is necessary to evaluate FinRL methods on multiple mainstream financial markets in different countries, such as US, Europe and China, to evaluate universality.

**Asset Type.** A financial asset is a liquid asset that derives its value from any contractual claim. Different asset types have different liquidity, trading rules and value models. It is necessary to evaluate FinRL methods on various financial asset types to evaluate universality.

**Time Scale.** We can evaluate FinRL methods on multiple trading scenarios with financial data on different time-scale (both coarse-grained and fine-grained). For instance, second-level data can be used for high frequency trading; minute-level data is suitable for intraday trading; day-level data can be applied for long-term trend trading.

**Rolling Window.** Due to the remarkable distribution shift in financial markets, researchers need to train and evaluate FinRL methods in a rolling time window, which means retrain or fine-tune RL models periodically to fit on current market status. Backtest with rolling window can evaluate the reliability of FinRL.

**t-SNE** is a statistical method for visualizing high-dimensional data by giving each datapoint a location in a two-dimensional map. First, t-SNE constructs a probability distribution over pairs of high-dimensional objects where similar objects are assigned a higher probability. Second, t-SNE defines a similar probability distribution over the points in the low-dimensional map, and it minimizes the KL divergence between the two distributions with respect to the locations of the points in the map. In FinRL, we use t-SNE to visualize financial datasets to show the relative position of them.

**Entropy** [20] is applied in finance to measure the amount of information give by observing the financial market. In a portfolio, it is defined as $H(\omega) = -exp\left(\sum_{i=1}^{n} \omega_i \log \omega_i\right)$, where $\omega = (\omega_1, \omega_2, ..., \omega_n)$ is a portfolio among $N$ financial assets. This measure reach a minimum value 1 if a portfolio is fully concentrated in one single asset and a maximum equal to $N$ that representation the equally weighted portfolio.

**Correlation.** As entropy ignore the correlation between different financial assets, effective number of bets (ENB) is proposed to remove the correlation while calculating entropy. We first define diversification distribution as $p_i(\omega) = \omega_{F_i}^2 \lambda_i^2 / \sum_{i=1}^{n} \omega_{F_i}^2 \lambda_i^2$. We formulate the covariance matrix of $N$ assets $\Sigma$ as $E'\Sigma E = \lambda$ where $\lambda$ is a diagonal matrix, $\lambda_i$ is the element of the diagonal matrix and $\omega_f = E^{-1}\omega$ where $\omega$ is the our original portfolios' weights. The effective number of bets is defined as: $ENB(\omega) = -exp\left(\sum_{i=1}^{n} p_i(\omega) \log p_i(\omega)\right)$, where $p_i(\omega)$ is the $i^{th}$ diversification distribution for the portfolios' weights $\omega$.

**Diversity Heatmap** is a visualization tool to demonstrate the diversity of investment decisions among different financial assets with heatmap [7]. The x-axis refers to the relative weight of each asset in the portfolio. The y-axis includes the results of different FinRL algorithms.

**Performance Profile** reports the score distribution of all runs across the 4 financial markets that are statistically unbiased, more robust to outliers and require fewer runs for lower uncertainty compared to point estimates such as mean. Performance profiles proposed herein visualize the empirical tail distribution function of a random score (higher curve is better), with point-wise confidence bands based on stratified bootstrap. A score distribution shows the fraction of runs above a certain normalized score that is an unbiased estimator of the underlying performance distribution.

**Variability** refers the variance of performance across different random seeds in RL [8]. As a high stake domain, it is important to test the variability, which is closely relevant to reliability.

**Rank Comparison**

In the rank distribution plot, the $i^{th}$ column shows the probability that a given method is assigned rank $i$ in the corresponding metrics, which provides a indication on the overall rank of FinRL methods.

## F   Experimental Setup of the Benchmarking Experiments

### F.1   Features

We generate 11 temporal features as shown in Table 4 to describe the stock markets following [3, 30]. $z_{open}$, $z_{high}$ and $z_{low}$ represent the relative values of the open, high, low prices relative to the close price at current time step, respectively. $z_{close}$ and $z_{adj\_close}$ represent the relative values of the closing and adjusted closing prices

Table 4: Features to describe the financial markets

| Features | Calculation Formula |
|---|---|
| $z_{open}, z_{high}, z_{low}$ $z_{close}$ | $z_{open} = open_t/close_t - 1$ $z_{close} = close_t/close_{t-1} - 1$ |
| $z_{d\_5}, z_{d\_10}, z_{d\_15}$ $z_{d\_20}, z_{d\_25}, z_{d\_30}$ | $z_{d\_5} = \dfrac{\sum_{i=0}^{4} close_{t-i}/5}{close_t} - 1$ |

compared with time step $t - 1$. $z_{dk}$ represents a long-term moving average of the adjusted close prices during the last $k$ time steps relative to the current close price. We apply z-score normalization on each feature.

### F.2   Algorithms

We pick the following 8 RLFT algorithms for the benchmarking experiments of portfolio management on US stocks:

- SARL [29] proposes a state-augmented RL framework, which leverages the price movement prediction as additional states, based on deterministic policy gradient [21] methods.
- EIIE [9] is considered as the first deep RLFT method with an ensemble of identical independent evaluators topology, a portfolio vector memory, and an online stochastic learning scheme.
- Investor-Imitator (IMIT) [1] is an RL approach that imitates behaviors of different types of investors using investor-specific reward functions with a set of logic descriptions.
- PPO[19] is a popular RL approach baseline that balances exploration and exploitation, making it stable and efficient.
- A2C [16] is an RL approach that combines both a policy function (actor) and a value function (critic) which achieves faster training through parallelization.
- DDPG [12] is an RL approach that leverages deep learning methods, actor-critic architecture, and stabilization techniques for continuous control problems
- DQN [15] is a milestone RL approach that combines Q-learning with deep learning. It is the foundation of many DRL methods.
- PG [25] is the foundation of Policy Gradient algorithms family.
- TD3 [4] is created based on DDPG, introducing several improvements that enhance robustness and efficiency.

## G   Extra Experiments

### G.1   Demonstrative Usage of the AutoML Component

To compare the results with or without automatic feature generation, we run PPO & SARL algorithms on DJ30 dataset for 5 epochs. We utilize the OpenFE library [32] to expand the candidate features using the original technical indicators of the dataset, and select the top-ranked 10 features as generated features to be appended to the dataset for the model training and testing. The quantitative results in Table 5 mirror the relative improvement of the total return using automatic feature generation compared to directly training by the base dataset.

Table 5: Comparison of results with or without the feature generation & selection

| Algorithm | Feature Generation & Selection | TR(%) | SR | VOL(%) | MDD (%) |
|---|---|---|---|---|---|
| PPO | × | 16.42±0.38 | 1.38±0.03 | 0.73±0.01 | 6.73±0.06 |
| | ✓ | **16.60±0.27** | **1.40±0.03** | **0.73±0.01** | **6.72±0.10** |
| SARL | × | 17.05±2.07 | 1.46±0.22 | 0.73±0.03 | 6.55±0.44 |
| | ✓ | **17.49±2.95** | **1.50±0.17** | **0.72±0.03** | **6.51±1.01** |

## G.2 Experiments on Three Time Windows with Mean and Standard Deviation

Table 6: US Stock 2021

| Metrics | A2C | DDPG | EIIE | IMIT | SARL | TD3 | PG | PPO |
|---|---|---|---|---|---|---|---|---|
| TR(%) | 16.54±0.13 | 17.88±0.12 | 14.55±0.1 | 12.4±4.98 | 13.75±1.94 | 17.88±0.08 | 16.42±0.2 | 16.52±0.2 |
| SR | 1.39±0.01 | 1.5±0.01 | 1.3±0.01 | 0.79±0.27 | 1.13±0.14 | 1.49±0.01 | 1.38±0.01 | 1.39±0.02 |
| CR | 2.38±0.01 | 2.58±0.02 | 2.11±0.02 | 1.25±0.54 | 1.95±0.38 | 2.57±0.02 | 2.34±0.05 | 2.39±0.03 |
| SoR | 2.0±0.01 | 2.14±0.01 | 1.8±0.01 | 1.11±0.36 | 1.64±0.17 | 2.13±0.0 | 1.98±0.02 | 1.99±0.02 |
| MDD(%) | 6.72±0.06 | 6.63±0.04 | 6.75±0.01 | 10.76±1.02 | 7.13±0.85 | 6.67±0.03 | 6.79±0.08 | 6.69±0.06 |
| VOL(%) | 0.72±0.0 | 0.72±0.0 | 0.72±0.0 | 1.05±0.06 | 0.78±0.04 | 0.73±0.0 | 0.72±0.0 | 0.72±0.0 |
| ENT | 2.15±0.01 | 2.66±0.12 | 3.37±0.01 | 1.79±0.07 | 2.79±0.08 | 2.63±0.22 | 2.15±0.02 | 2.13±0.04 |
| ENB | 1.39±0.02 | 1.33±0.07 | 1.11±0.0 | 1.8±0.06 | 1.1±0.04 | 1.23±0.08 | 1.38±0.02 | 1.39±0.03 |

Table 7: US Stock 2020

| Metrics | A2C | DDPG | EIIE | IMIT | SARL | TD3 | PG | PPO |
|---|---|---|---|---|---|---|---|---|
| TR(%) | 7.78±0.26 | 8.68±0.15 | 9.69±0.14 | -13.69±4.06 | 11.48±2.14 | 8.77±0.12 | 7.4±0.24 | 7.1±0.38 |
| SR | 0.39±0.01 | 0.41±0.01 | 0.45±0.01 | -0.04±0.09 | 0.49±0.06 | 0.42±0.01 | 0.38±0.01 | 0.37±0.01 |
| CR | 0.41±0.01 | 0.44±0.01 | 0.48±0.01 | -0.04±0.1 | 0.53±0.06 | 0.44±0.01 | 0.40±0.01 | 0.39±0.01 |
| SoR | 0.47±0.01 | 0.50±0.01 | 0.54±0.01 | -0.05±0.12 | 0.59±0.07 | 0.5±0.01 | 0.45±0.01 | 0.45±0.01 |
| MDD(%) | 32.44±0.14 | 32.37±0.09 | 30.86±0.19 | 45.23±3.13 | 32.23±2.34 | 32.36±0.05 | 32.49±0.11 | 32.49±0.13 |
| VOL(%) | 2.16±0.01 | 2.16±0.01 | 2.13±0.01 | 3.19±0.04 | 2.21±0.13 | 2.15±0.01 | 2.16±0.01 | 2.16±0.01 |
| ENT | 2.16±0.01 | 2.7±0.08 | 3.37±0.01 | 1.5±0.03 | 2.86±0.06 | 2.61±0.15 | 2.13±0.02 | 2.14±0.02 |
| ENB | 1.04±0.01 | 1.02±0.01 | 1.02±0.01 | 1.17±0.01 | 1.02±0.02 | 1.03±0.01 | 1.04±0.01 | 1.04±0.01 |

Table 8: US Stock 2019

| Metrics | A2C | DDPG | EIIE | IMIT | SARL | TD3 | PG | PPO |
|---|---|---|---|---|---|---|---|---|
| TR(%) | 20.95±0.21 | 22.51±0.12 | 19.11±0.04 | -2.84±8.44 | 19.96±4.19 | 22.57±0.09 | 20.9±0.08 | 20.99±0.09 |
| SR | 1.82±0.02 | 1.95±0.01 | 1.76±0.01 | -0.04±0.44 | 1.7±0.28 | 1.95±0.01 | 1.82±0.01 | 1.83±0.01 |
| CR | 3.2±0.04 | 3.48±0.01 | 3.14±0.01 | 0.04±0.42 | 2.75±0.64 | 3.51±0.02 | 3.21±0.02 | 3.21±0.03 |
| SoR | 2.22±0.02 | 2.37±0.01 | 2.11±0.01 | -0.04±0.62 | 2.18±0.34 | 2.37±0.01 | 2.21±0.01 | 2.23±0.01 |
| MDD(%) | 6.14±0.04 | 6.0±0.03 | 5.76±0.02 | 22.83±5.19 | 6.91±0.39 | 5.98±0.03 | 6.1±0.03 | 6.13±0.05 |
| VOL(%) | 0.69±0.01 | 0.69±0.01 | 0.68±0.01 | 1.29±0.08 | 0.72±0.03 | 0.69±0.01 | 0.69±0.01 | 0.69±0.01 |
| ENT | 2.17±0.03 | 2.64±0.18 | 3.37±0.01 | 1.09±0.1 | 2.62±0.21 | 2.59±0.12 | 2.16±0.01 | 2.14±0.01 |
| ENB | 1.24±0.02 | 1.16±0.03 | 1.04±0.01 | 2.02±0.08 | 1.13±0.04 | 1.15±0.06 | 1.25±0.01 | 1.27±0.01 |

## G.3 Convergence Curve of Different Algorithms

## H Accessibility, Maintenance, License and Future Plan

TradeMaster is an open-source project on GitHub under the Apache 2.0 license for research and educational purpose. All source code, tutorials, scripts, benchmarks and supporting documentations are available on the GitHub repository: https://github.com/TradeMaster-NTU/TradeMaster.

TradeMaster has been actively maintained by the AMI group at Nanyang Technological University. Through GitHub, we keep including new features, improving code structures, merging pull requests, fixing bugs and responding to issues. Contributions from other researchers and investors are highly welcomed. In addition, we post a summary of change logs in the README web page to help users quickly catch up new features in the code repository. We hope TradeMaster can facilitate the development of RLFT field and emerges a open-source Fintech community at the intersection and AI and finance. We plan to keep improving TradeMaster with more financial markets, trading scenarios, RL algorithms and support of real-world trading.

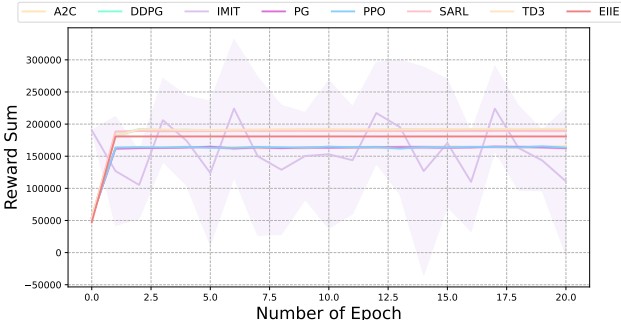

Figure 5: Convergence curve of different algorithms for training. All 8 algorithms converge after training epochs, where each epoch goes through 10 years of stock market history.