# OpenReview forum: "TradeMaster: A Holistic Quantitative Trading Platform Empowered by Reinforcement Learning"
_NeurIPS.cc/2023/Track/Datasets_and_Benchmarks — NeurIPS 2023 Datasets and Benchmarks Poster_

### Official Review · Reviewer_fVic · 2023-07-06
**Good software and open-source library, but not a good benchmark design, and lacks insightful machine learning findings.**

**Rating:** 6
**Confidence:** 4
**Clarity:** Yes.

**Strengths:**

1. As pointed out in the challenges, the work of implementating details occupies >90% in the whole pipeline, and <10% of work on core technical questions. This RLFT platform will fill the gap between laboratory financial algorithms and real-world financial markets, providing users a useful toolkit to practically and conveniently evaluate their new RL techniques and models.
2. This platform provides a convenient tool to fairly compare different RLFT techniques with the same evaluation conditions, including 13 real-world financial datasets and 6 mainstream quantitative trading tasks.
3. As a benchmark, this paper provides comprehensive comparison of 15 popular RLFT algorithms.
4. Considering the impact of extreme market conditions makes the benchmark more comprehensive.

**Additional Feedback:**

See Opportunities For Improvement and limitations.

**Correctness:**

As a benchmark, this paper does not provide enough experiments, including hyper-parameters, training epochs, and standard deviations.

**Documentation:**

Good and comprehensive.

**Ethics:**

No ethical cconcerns.

**Limitations:**

1. As a software, TradeMaster is similar to [4,5].
2. As a benchmark, although there are many baseline RL algorithms evaluated, this paper does not provide enough experiments, including hyper-parameters, training epochs, and standard deviations.

[4] Xiao Yang, Weiqing Liu, Dong Zhou, Jiang Bian, and Tie-Yan Liu. Qlib: An ai-oriented quantitative investment platform. arXiv preprint arXiv:2009.11189, 2020.
[5] Xiao-Yang Liu, Hongyang Yang, Jiechao Gao, and Christina Dan Wang. Finrl: Deep reinforcement learning framework to automate trading in quantitative finance. In ACM International Conference on AI in Finance, pages 1–9, 2021.

**Opportunities For Improvement:**

1. In the training setup, each algorithm is trained for 10 epochs. I'm wondering whether 10 epochs are enough for convergence. Could authors answer this question and offer convergence curves to support?
2. For hyperparameters, are default choices of TradeMaster suitable for each algorithm?
3. In the Chapter 5, the demonstrative code shows that whole pipeline of TradeMaster. However, could authors demonstrate more about how it is ``simple, clean and easy to extend''? Specifically, how to develope and seamlessly plug new algorithms into the TradeMaster, without considering other implementation details.

**Relation To Prior Work:**

Prior works[1,2] provide DataOps paradigm, traditional finance open-source libraries [3], AI-oriented quantitative investment platforms [4,5]. TradeMaster belongs to the same kind of [4,5]. It seems that [4,5] have similar software features of TradeMaster.

[1] Julian Ereth. Dataops-towards a definition. LWDA, 2191:104–112, 2018
[2] Xiao-Yang Liu, Ziyi Xia, Jingyang Rui, Jiechao Gao, Hongyang Yang, Ming Zhu, Christina Dan Wang, Zhaoran Wang, and Jian Guo. FinRL-Meta: Market environments and benchmarks for data-driven financial reinforcement learning. In Advances in Neural Information Processing Systems (Datasets and Benchmarks Track), 2022.
[3] N Firth. Why use quantlib. URL http://www. maths. ox. ac. uk/ firth/research/quantlib. pdf, 2004.
[4] Xiao Yang, Weiqing Liu, Dong Zhou, Jiang Bian, and Tie-Yan Liu. Qlib: An ai-oriented quantitative investment platform. arXiv preprint arXiv:2009.11189, 2020.
[5] Xiao-Yang Liu, Hongyang Yang, Jiechao Gao, and Christina Dan Wang. Finrl: Deep reinforcement learning framework to automate trading in quantitative finance. In ACM International Conference on AI in Finance, pages 1–9, 2021.

**Summary And Contributions:**

It is challenging to deploy reinforcement learning (RL) methods into real-world financial markets due to the highly composite nature of this domain. This paper introduces TradeMaster, a holistic open-source RLFT platform that serves as a software toolkit, empirical benchmark, and user interface, to provide infrastructures for transparent and reproducible RLFT research and facilitate their real-world deployment with industry impact.

---

> ### Author Response · Authors · 2023-08-21
> **Response to Reviewer fVic**
>
> We thank the reviewer for the valuable feedback and suggestions. We have revised the paper and respond to questions as follows:
>
> **Q1. There are lots of RL for QT projects out there. The authors didn’t clearly state why TradeMaster is better than the other.**
>
> We have different opinions on this point with the reviewer. TradeMaster is the first holistic platform for RL-based quantitative trading, which is not similar to any other existing platform. We add a paragraph (line 337-349) to point out limitations of existing work and uniqueness of TradeMaster: i) while existing platforms claims support of various markets, they only offer links to many data providers that users have to write scripts to acquire data flexibly. TradeMaster prepares 13 well-processed datasets and serve them on Google Drive to be downloaded directly. ii) TradeMaster covers a wide range of trading scenarios with realistic simulation under practical constraints. As far as we know, many scenarios (e.g., intraday trading and market making) are not covered by existing platforms. iii) TradeMaster provides high-quality implementations of 9 classic RL algorithms and 7 RL for trading algorithms, where about half of them is not covered by existing platforms. iv) TradeMaster includes a series of evaluation measures and visualization toolkits to provide a systematic evaluation instead of pure financial metrics in existing platforms. v) We build a GUI interface on the official TradeMaster website for users without in-depth coding skills to explore the potential of RLFT and offer a Colab version for convenient cloud training. vi) We include an AutoML module for automatic feature selection and hyperparameter tuning to further improve usability.
>
> In addition, we provide a comparison of TradeMaster and existing trading platforms in the following table, where # indicates "the number of".
> | Platform    | # financial market | # RLFT algorithms | # trading scenarios | # evaluation metrics | # plot toolkits | GUI interface | Colab version | AutoML |
> | -------- | ------- | ------- | ------- | ------- | ------- | ------- | ------- | ------- |
> | QLib [1]  |  2  | 1 | 1 | 4 | - | $\times$ | $\times$ | $\times$ |
> | FinRL [2] | 2  |  8 | 3 | 5 | -   | $\times$ | $\surd$ | $\times$ |
> | TradeMaster (ours)    | 13 | 16 | 6 | 10 | 6 | $\surd$ | $\surd$ | $\surd$ |
>
> [1] Yang et al. Qlib: An AI-oriented quantitative investment platform. arxiv. 2020.
>
> [2] Liu et al. Finrl: Deep reinforcement learning framework to automate trading in quantitative finance. International Conference on AI in Finance. 2021.
>
> ---
> **Q2. In the training setup, each algorithm is trained for 10 epochs. I’m wondering whether 10 epochs are enough for convergence. Please offer convergence curves to support.**
>
> We would like to clarify on this problem with the following three perspectives:
> * Unlike classic RL environments such as Atari, there is significantly temporal distribution shift due to the change of market status. There is a strong tendency to overfit with poor generalization performance on unseen market when training RL algorithms for too many epochs on financial data [3].
> * Training for 10 epochs is intuitively enough in this case because each epoch goes through about 10 years of market data while training, which should be long enough for RL agents to learn on different types of financial markets.
> * To further support the above points, we include the training curves of 8 algorithms in Appendix F.3. As shown in the figure, all 8 algorithms converge after training for 10 epochs.
>
> [3] Prado et al. Advances in financial machine learning. John Wiley. 2018.

---

> ### Author Response · Authors · 2023-08-21
> **Response to Reviewer fVic**
>
> **Q3. For hyperparameters, are default choices of TradeMaster suitable for each algorithm?**
>
> We first select the values of hyperparamters following two conditions: i) if there are authors’ official or open-source RLFT library implementations, we apply the same hyperparameters for a fair comparison since they are tuned in financial domain. This condition applies for A2C, DDPG, TD3, PG, PPO and SARL. ii) if there are no publicly available implementations, we reimplement the algorithms and try our best to maintain consistency based on the original papers. This applies for EIIE and IMIT. Later on, we apply grid search on several key RL hyperparameters based on the TradeMaster codebase to further improve the performance. Specificially, we try batch size in list [256, 512, 1024], hidden size in range [64, 128, 256] and learning rate in [3$e^{-4}$,5$e^{-4}$,7$e^{-4}$,9$e^{-4}$] for both actor and critic. Adam is used as the optimizer. More details on the hyperparamters are available in the TradeMaster GitHub repository. We include this discussion in line 243.
>
> ---
> **Q4. In Chapter 5, the demonstrative code shows the whole pipeline of TradeMaster. However, could authors demonstrate more about how it is “simple, clean and easy to extend”? Specifically, how to develop and seamlessly plug new algorithms into the TradeMaster, without considering other implementation details.**
>
> The software architecture of TradeMaster is simple and clean with proper encapsulation, which enable users to add new features by changing very limited amount of code. We provide two concrete examples as follows:
> * When users would like to develop a new RL algorithm, they can create a new file under the TradeMaster/agents folder. Later on, they can implement the algorithm based on their design. By making sure the input and output is aligned with the requirement, the algorithm can work immediately without changes on other parts.
> * When users would like to train RL algorithms on other financial markets, they can do it directly by changing the corresponding directory in the config file and adding a new csv data file containing the required original features (e.g., open, high, low, close).
>
> ---
>
> **Q5. As a benchmark, although there are many baseline RL algorithms evaluation, this paper deos not provide enough experiments, including hyper-parameters, training epochs and standard deviations.**
>
> We address this concern with the following two points:
> * As required by the reviewer, we put the experimental results with mean and standard deviation of the 5 random seeds in Appendix F.2. We also include details of experimental setup, hyperparameter selection and training epochs (line 243).
> * Note that the contribution of TradeMaster is not limited to a reproducible benchmark for RL-based quantitative trading. It also serve as a holistic software platform and user interface. Due to page limit, we can not include too much experimental results in the main text. We include extra experiments in Appendix F and the Github repository (https://github.com/TradeMaster-NTU/TradeMaster/tree/1.0.0/tutorial).
>
>
> ---
>
> *We would appreciate it if reviewers can confirm that their concerns had been addressed. if so, reconsider their assessment. We’d be happy to engage in further discussions.*

---

> > ### Comment · Reviewer_fVic · 2023-08-29
> > **Thanks for your responses**
> >
> > Thanks for your responses. Most of my questions are addressed. I have checked the newly provided convergence curve, most algorithms converge in 2 epochs except IMIT. Could your provide some explanations about this? And what does the shadow color in Figure 5 in Appendix mean?

---

> > > ### Author Response · Authors · 2023-08-29
> > > **Responses to Reviewer fVic**
> > >
> > > Dear reviewer fVic,
> > >
> > > Thank you for the timely feedback. We respond to your questions as follows:
> > >
> > > **Q1. I have checked the newly provided convergence curve, most algorithms converge in 2 epochs except IMIT. Could you provide some explanations about this?**
> > >
> > > We would like to clarify on this with the following four points:
> > > * Two epochs go through about 20 years of market data, which is long enough for most RL algorithms to learn on investment decision making under different market status.
> > > * Unlike most algorithms that train RL agents to make investment decisions directly, IMIT [1] focuses on imitating the trading behaviors of different experts with two stages. In stage one, IMIT trains multiple supervised learning trading experts through optimizing various financial metrics (e.g., return rate and Sharpe ratio). In stage two, IMIT trains RL agents to construct profitable portfolios by dynamically picking pre-trained experts.
> > > * As stated in the previous point, the goal of IMIT's RL training is to dynamically pick different pre-trained experts under suitable market status. This is a hard task because the advantage difference among various experts can be unobvious and usually require lots of efforts to identify. For instance, one expert with the best return rate can also achieve good Sharpe ratio at the same time because these two metrics are correlated. As a result, it takes IMIT more epochs to converge due to the difficulty of the task.
> > > * The start point (performance on epoch 0) of IMIT is higher than other algorithms because all pre-trained supervised learning experts already have reasonably good performance comparing to other algorithms that start from random portfolios.
> > >
> > > [1] Ding et al. Investor-imitator: a framework for trading knowledge extraction. KDD. 2018.
> > >
> > > ---
> > >
> > > **Q2. What does the shadow colour in Figure 5 of Appendix mean?**
> > >
> > > The shadow colour indicates the standard deviation of training curves across 5 random seeds.
> > >
> > > ---
> > > *We hope the above clarification address your concerns. We are happy to engage in further discussion!*

---

> > > > ### Comment · Reviewer_fVic · 2023-08-29
> > > > **Thanks for your response**
> > > >
> > > > I have no further questions. I'd like to increase my scores according to the contribution of this paper and efforts during rebuttal.

---

> > > > > ### Author Response · Authors · 2023-08-29
> > > > > **Thank you!**
> > > > >
> > > > > Dear reviewer fVic,
> > > > >
> > > > > Thanks again for your valuable feedback and suggestions. We are very happy that our responses address your concerns. Thank you very much for raising the score.
> > > > >
> > > > > Best regards,
> > > > >
> > > > > Authors of Submission 141

---

> ### Author Response · Authors · 2023-08-24
> **Does our response address your concerns?**
>
> Dear reviewer fVic,
>
> Thanks again for your valuable feedback and insightful suggestions. During the rebuttal phase, we carefully go through them, revise the paper and find that there might be some misunderstandings. To address the concerns, we highlight the uniqueness of TradeMaster comparing to other platforms. We also provide additional details on hyperparameter selection, experimental setup and training epochs. Furthermore, we add experiments of the learning curve to show the convergence and results table with standard deviation as additional information for benchmark. Finally, we offer concrete examples to show how to easily extend a customized version of TradeMaster based on users' personal requirements.
>
> Could you please take some valuable time to have a look at our responses? We would really appreciate it if you can confirm that the concerns have been addressed and, if so, reconsider the assessment. We are happy to engage in further discussions.
>
> Best regards,
>
> Authors of Submission 141

---

> ### Author Response · Authors · 2023-08-28
> **Does our response address your concerns? (Deadline on Aug 30)**
>
> Dear reviewer fVic,
>
> Thanks again for your valuable feedback and insightful suggestions. During the rebuttal phase, we carefully go through them, revise the paper and find that there might be some misunderstandings. To address the concerns, we highlight the uniqueness of TradeMaster comparing to other platforms. We also provide additional details on hyperparameter selection, experimental setup and training epochs. Furthermore, we add experiments of the learning curve to show the convergence and results table with standard deviation as additional information for benchmark. Finally, we offer concrete examples to show how to easily extend a customized version of TradeMaster based on users' personal requirements.
>
> As it is close to the deadline, could you please take some valuable time to have a look at our responses? We would really appreciate it if you can confirm that the concerns have been addressed and, if so, reconsider the assessment. We are happy to engage in further discussions.
>
> Best regards,
>
> Authors of Submission 141

---

### Official Review · Reviewer_k4uA · 2023-07-09
**Very Interesting Work!**

**Rating:** 7
**Confidence:** 3
**Correctness:** No related issues are identified.
**Clarity:** The paper is informative and well-wri…

**Strengths:**

Significance of the contribution: The aim of creating an all-encompassing platform is notable, which tries to provid a one-stop solution for academic researchers and industry practitioners alike. The developed platform is a valuable toolset for RLFT applications.

Quality of the research: Quite impressive, demonstrating a substantial amount of work. This reflects the significant effort and dedication invested by the author team into the project.

Relevance: Yes, relevant to the Datasets and Benchmarks Track

Clarity of paper: The paper is well-structured and articulate, with clear exposition of the TradeMaster platform and its functionalities.


**Additional Feedback:**

Detailed feedback comments can be seen in other sections.

**Documentation:**

The provided instructions are satisfactory.

**Ethics:**

No ethical concerns are identified.

**Limitations:**

Platforms like TradeMaster possess significant potential to advance research in the relevant community. However, a notable challenge is attracting a broader pool of researchers to leverage this platform effectively. It would be insightful to delve into the authors' perspectives and analyses regarding strategies to overcome this bottleneck and enhance the platform's impact and reach.

**Opportunities For Improvement:**

Although TradeMaster has made significant progress in addressing crucial aspects of RLFT, it is important to acknowledge that the complexities of financial markets and the dynamic nature of RL algorithms pose inherent challenges. To fully appreciate the platform, further exploration into how TradeMaster handles these complexities and its potential scalability to accommodate diverse research needs would be beneficial. Additionally, emphasizing the ease of use is crucial for the practicality and popularity of such platforms. The authors can discuss their plans for enhancing user-friendliness and accessibility, which would enable a broader user base to leverage TradeMaster effectively. By addressing these considerations, the platform can realize its true value in simplifying and advancing RLFT applications in financial trading.

**Relation To Prior Work:**

The authors have adequately discussed how their work differs from previous contributions.

**Summary And Contributions:**

This paper introduces TradeMaster, an open-source platform that aims to tackle the persistent challenges associated with using reinforcement learning in financial trading (RLFT). The contributions of TradeMaster are multi-fold. Firstly, it provides a comprehensive software toolkit that includes real-world financial datasets and popular RLFT algorithms, facilitating the practical implementation of RLFT in trading scenarios. Secondly, it establishes an empirical benchmark for comparing RLFT algorithms, enabling researchers and practitioners to evaluate and compare the performance of different approaches. Lastly, TradeMaster offers user-friendly interfaces with the support of AutoML, enhancing the accessibility and usability of the platform for a diverse range of users. By leveraging TradeMaster, the application of RLFT in financial trading could be simplified and enriched, fostering advancements in the field.

---

> ### Author Response · Authors · 2023-08-21
> **Response to Reviewer k4uA**
>
> We thank the reviewer for the valuable feedback and suggestions. We have revised the paper and respond to questions as follows:
>
> **Q1. It’s beneficial to explore how TradeMaster handles the complexities of financial markets and the dynamic nature of RL algorithms and its potential scalability to accommodate diverse research needs.**
>
> * To handle the complexities of financial markets and RL, TradeMaster offers efficient open-source implementations of the whole RLFT workflow to enable fast development and test for different users. In addition, we host TradeMaster as a benchmark to provide reproducible comparison on different algorithms.
> * We highlight the potential impact of TradeMaster for different communities (line 351-358). For RLFT researchers, they can rapidly implement and compare their own methods with a focus on real scientific problems instead of engineering details. For RL researchers, TradeMaster introduces many challenging trading scenarios for the test of novel RL algorithms in financial markets. For finance researchers and individual investors, they can have a taste on RL-based trading methods without in-depth knowledge of AI and coding. For professional trading firms, TradeMaster can serve as the code base to enhance their internal trading systems with advanced RL techniques.
>
> ---
> **Q2. It is crucial to emphasize the ease of use for TradeMaster. The authors can discuss their plans for enhancing user-friendliness and accessibility.**
>
> We highlight our current efforts on enhancing user experience and accessibility as follows:
> * We host the TradeMaster project on GitHub (over 750 stars) with an "what is new" table to show our updates on the platform. All related resources (e.g., datasets, website and software documents) are publicly available, where users can flexibly use under the Apache 2.0 License.
> * We develop an online GUI interfaces (details in Appendix B) on the TradeMaster website (http://trademaster.ai/)  for users without in-depth coding knowledges to train and test RL algorithms on different trading tasks through interacting with the graphical interface.
> * TradeMaster supports automatic feature selection and hyperparameter tuning to improve its accessibility for users unfamiliar with RL.
> * A series of evaluation and visualization tools are provided in TradeMaster to help users evaluate and analyse the performance of RL algorithms.
>
> In the future, we will continuously update and maintain the project based on the feedback from different communities to improve its user experiences and accessibility.
>
> ---
> **Q3. A notable challenge for TradeMaster is attracting a broader pool of researchers to leverage the platform effectively. It would be insightful to delve into the authors' perspectives and analyses regarding strategies to overcome this bottleneck and enhance the platform's impact.**
>
> We discuss on our thoughts and efforts on attracting more users and enhancing the platform's impact as follows:
> * TradeMaster can be installed and used on multiple operating systems including Windows, Linux and MacOS.
> * To fit for the requirement of different users, TradeMaster offers lots of versions including functional API,a Python package, an online web service with a graphical user interface (GUI), multiple Jupyter Notebook tutorials and a cloud version using Colab.
> * We closely collaborate with several Singapore-based trading firms to explore RL’s potential in real-world trading applications with industry-level impact.
> * We provide detailed software documents of TradeMaster (https://trademaster.readthedocs.io/en/latest/) to help different users get familiar with the project.
> * The TradeMaster team has been invited to give talks and tutorial about RL-based trading and usage of TradeMaster platform on several top AI conferences (e.g., AAAI23, KDD23, IJCAI23).
> * We have organized an AAAI summer symposium (https://sites.google.com/view/aaai23-ai4fintech) based on TradeMaster, which attracts 58 people from 5 countries to communicate on AI4Fintech.
>
> ---
> *We would appreciate it if reviewers can confirm that their concerns had been addressed. We’d be happy to engage in further discussions.*

---

### Official Review · Reviewer_DHLR · 2023-07-20
**Review for "TradeMaster: A Holistic Quantitative Trading Platform Empowered by Reinforcement Learning"**

**Rating:** 6
**Confidence:** 3
**Correctness:** For the best knowledge of reviewer, t…
**Clarity:** The paper is well written.

**Strengths:**

The paper is clearly written in terms of the challenges and contributions mentioned. It seems that the authors provide a well-constructed platform to implement reinforcement learning in financial data. To demonstrate the platform, the authors provide a clear review about the basic concepts in RL, which helps readers to better understand the scope of the paper.

**Additional Feedback:**

Please see the comments above.

**Documentation:**

Yes.

**Ethics:**

No.

**Limitations:**

Please see in above.

**Opportunities For Improvement:**

1. The foundations of RL replies on Markov assumptions in Markov Decision Making Process (MDP). Although there are several previous literature that focus on proposing RL methods for quantitative data, it is questionable whether the data satisfies the Markov stationary assumptions. In practice, the finance data can be highly fluctuate and non-stationary. Hence, although the data generation is related to MDP, the implementation of RL methods for it can be challenging.
2. For the experimental results, are the included algorithms specifically designed for financial data? Are there any reasons about why some methods can/cannot perform good in various datasets?

**Relation To Prior Work:**

The paper has some discussions about the related literature of RL in financial data.

**Summary And Contributions:**

The deployment of reinforcement learning (RL) methods in real-world financial markets remains challenges. The authors present the TradeMaster, an open-source Reinforcement Learning for Finance and Trading (RLFT) platform that serves as a software toolkit, empirical benchmark, and user interface. The platform can help establish infrastructures that enable transparent and reproducible RLFT research while facilitating practical application with significant industry impact.

---

> ### Author Response · Authors · 2023-08-21
> **Response to Reviewer DHLR**
>
> We thank the reviewer for the valuable feedback and suggestions. We have revised the paper and respond to questions as follows:
>
> **Q1. The foundations of RL rely on the Markov assumptions of MDP. Although there are literatures focusing on RL methods for trading, it is questionable whether the financial data satisfies the Markov stationary assumptions. As a result, the implementation of RL methods for this domain can be challenging.**
>
> We would like to clarify on this concern from the following 5 perspectives:
> * For the Markov assumptions on financial trading tasks, we typically assume that the market status of next time step only relying on what happened in the near past. Based on the efficient market hypothesis [1], the asset price reflects all available information within an extremely short time window, which holds the Markov assumptions.
> * We would like to highlight that the Markov property holds better in the high frequency regime since the impact of macro-level information (e.g., government policy) is tiny in the short time scale.
> * In TradeMaster, we try to enhance the Markov property through offering high-quality implementation of RL environments on multimodality data to cover as much market information as possible.
> * Many industry trading firms (e.g., ARK Invest and Lingjun Investment) have shown interest on exploring the potential of RL in trading applications.
> * For academic research, there has been many works focusing on RL methods for various trading tasks with great performance [2], which further demonstrate the existence of Markov property.
>
> [1] Fama et al. Efficient capital markets: A review of theory and empirical work. The Journal of Finance. 1970.
>
> [2] Sun et al. Reinforcement learning for quantitative trading. ACM Transactions on Intelligent Systems and Technology. 2023.
>
> ---
> **Q2. For the experimental results, are the included algorithms specifically designed for financial data?**
>
> We include descriptions on the baseline selection (line 243). Specifically, there are 3 RL algorithms (e.g., EIIE, IMIT and SARL) designed for financial data among our 8 baselines. More details are available in line 179-186.
>
> ---
> **Q3. Are there any reasons about why some methods can/cannot perform good in various datasets?**
>
> Besides our existing discussion in line 228-235, we include more contents (line 235-242) to shed light on the analysis of experiment results. Generally speaking, no existing algorithm achieves dominating results on all financial metrics and there are still vast opportunities to improve in this filed. In addition, we observe that classic RL algorithms (e.g., DDPG and TD3) still outperform RL-based trading algorithms, when proper hyperparameters are applied, in terms of profit-related metrics (e.g., TR, SR, CR and SoR). At the same time, RLFT algorithms (e.g., EIIE, IMIT and SARL) achieve portfolios with better risk-control and diversity, which demonstrate the effectiveness of the risk-related components. We hope the reproducible benchmark results from TradeMaster can help researchers easily notice the relative strength of existing algorithms and inspire the design of new algorithms.
>
> ---
> *We would appreciate it if reviewers can confirm that their concerns had been addressed. We’d be happy to engage in further discussions.*

---

### Official Review · Reviewer_Vtxm · 2023-07-22
**More works need to be done**

**Rating:** 7
**Confidence:** 5
**Correctness:** Yes
**Clarity:** No

**Strengths:**

This project is open-sourced with functional API,a Python package, an online web service with a graphical user interface (GUI), multiple Jupyter Notebook tutorials and a cloud version using Colab. The setup is good for the initial stage of QT

**Additional Feedback:**

I think the authors should compare other open-sourced RL for QT projects.

**Documentation:**

Yes

**Limitations:**

I think the paper can do a lot better as an open-sourced project. There are lots of RL for QT projects out there. The authors didn't clearly state why RLFT is better than the other.

**Opportunities For Improvement:**

I think this paper/project has a lot to do.


One of the main problems for RL in QT is overhead/time. I would like to see that under the same hardware, for example, 3090 that is used in this paper, compared with other open-sourced work, how the performance will be?
The MDP steps seem really unclear to me. For example, it's obvious that for different tasks, the Action space/state space is different, the author only briefly states it, not in detail. And figure 2 is a general figure, not a specific figure showing how the reward will be facing different actions, etc.

**Relation To Prior Work:**

No

**Summary And Contributions:**

The authors introduce TradeMaster, a holistic open-source RLFT platform that serves as a i) software toolkit, ii) empirical benchmark, and iii) user interface.

RLFT includes 13 real-world financial datasets, high-fidelity RL environments for 6 mainstream quantitative trading tasks, 15 popular RLFT algorithms, and dozens of tools for systematic evaluation and visualization.

---

> ### Author Response · Authors · 2023-08-21
> **Response to Reviewer Vtxm**
>
> We thank the reviewer for the valuable feedback and suggestions. We have revised the paper and respond to questions as follows:
>
> **Q1. I think the paper can do a lot better as an open-source project.**
>
> We have different opinions on this point with the reviewer. We believe TradeMaster has already benefited many communities as an open-source project while the current version is not a panacea. We would like to highlight our efforts and achievements on building TradeMaster as an ecosystem:
> * The Github repository of TradeMaster (https://github.com/TradeMaster-NTU) attracts many users from different communities with over 750 stars and 170 folks.
> * We build an official website of TradeMaster (http://trademaster.ai/) including related information of the project and a GUI interface for training/testing RLFT algorithms.
> * The TradeMaster team has been invited to give talks and tutorials on several top AI conferences (e.g., AAAI23, KDD23, IJCAI23).
> * We provide detailed software documents of TradeMaster (https://trademaster.readthedocs.io/en/latest/) to make it more accessible for different users.
> * We have organized an AAAI summer symposium (https://sites.google.com/view/aaai23-ai4fintech) based on TradeMaster, which attracts 58 people from 5 countries to communicate on AI4Fintech.
> * We closely collaborate with several Singapore-based trading firms to explore RL’s potential in real-world trading applications with industry-level impact.
>
> ---
> **Q2. There are lots of RL for QT projects out there. The authors didn’t clearly state why TradeMaster is better than the other.**
>
> To address this concern, we add a paragraph (line 337-349) to point out limitations of existing work and uniqueness of TradeMaster: i) while existing platforms claims support of various markets, they only offer links to many data providers that users have to write scripts to acquire data flexibly. TradeMaster prepares 13 well-processed datasets and serve them on Google Drive to be downloaded directly. ii) TradeMaster covers a wide range of trading scenarios with realistic simulation under practical constraints. As far as we know, many scenarios (e.g., intraday trading and market making) are not covered by existing platforms. iii) TradeMaster provides high-quality implementations of 9 classic RL algorithms and 7 RL for trading algorithms, where about half of them is not covered by existing platforms. iv) TradeMaster includes a series of evaluation measures and visualization toolkits to provide a systematic evaluation instead of pure financial metrics in existing platforms. v) We build a GUI interface on the official TradeMaster website for users without in-depth coding skills to explore the potential of RLFT and offer a Colab version for convenient cloud training. vi) We include an AutoML module for automatic feature selection and hyperparameter tuning to further improve usability.
>
> In addition, we provide a comparison of TradeMaster and existing trading platforms in the following table, where # indicates "the number of".
> | Platform    | # financial market | # RLFT algorithms | # trading scenarios | # evaluation metrics | # plot toolkits | GUI interface | Colab version | AutoML |
> | -------- | ------- | ------- | ------- | ------- | ------- | ------- | ------- | ------- |
> | QLib [1]  |  2  | 1 | 1 | 4 | - | $\times$ | $\times$ | $\times$ |
> | FinRL [2] | 2  |  8 | 3 | 5 | -   | $\times$ | $\surd$ | $\times$ |
> | TradeMaster (ours)    | 13 | 16 | 6 | 10 | 6 | $\surd$ | $\surd$ | $\surd$ |
> |
>
>
> [1] Yang et al. Qlib: An AI-oriented quantitative investment platform. arxiv. 2020.
>
> [2] Liu et al. Finrl: Deep reinforcement learning framework to automate trading in quantitative finance. International Conference on AI in Finance. 2021.

---

> > ### Author Response · Authors · 2023-08-28
> > **Does our response address your concerns? (Deadline on Aug 30)**
> >
> > Dear reviewer Vtxm,
> >
> > Thanks again for your valuable feedback and insightful suggestions. During the rebuttal phase, we carefully go through them, revise the paper and find that there might be some misunderstandings. To address the concerns, we highlight the uniqueness of TradeMaster and point out our efforts and achievements on serving TradeMaster as an open-source project. We also include discussion on the MDP formulation of RL for financial trading with concrete examples and talk about our thoughts on improving the algorithms efficiency.
> >
> > As it is close to the rebuttal deadline, could you please take some valuable time to have a look at our responses? We would really appreciate it if you can confirm that the concerns have been addressed and, if so, reconsider the assessment. We are happy to engage in further discussions.
> >
> > Best regards,
> >
> > Authors of Submission 141

---

> > ### Comment · Reviewer_Vtxm · 2023-08-31
> > **I think FinRL has a Colab version...**
> >
> > This is only for pure discussion purposes:
> >
> > I think FinRL does have a Colab version. The ICAIF 2021 paper is kind of out of date. It's a bit unfair to compare the year 2023 work with the previous version of other's work. But I agree that both TradeMaster and FinRL have their own advantages.
> >
> > The reason that I said "I think the paper can do a lot better as an open-source project", is because, for the AI4Finance community, people have really different backgrounds. You should put some effort on how to explain the benefit of RL for Finance to the community. Again, I said, "Can do a better job", I didn't say trademaster is bad.

---

> > > ### Author Response · Authors · 2023-08-31
> > > **Thank you very much for the reply!**
> > >
> > > Dear reviewer Vtxm,
> > >
> > > Thanks for your timely feedback and valuable suggestions. We are very happy that our responses address your concerns. Thank you very much for raising the score! As for the new discussion, our responses are as follows:
> > >
> > > **Q1. I think FinRL does have a Colab version. It's a bit unfair to compare the year 2023 work with the previous version of other's work. But I agree that both TradeMaster and FinRL have their own advantages.**
> > >
> > > Thank you very much for the notice. We have fixed it in the comparison table. We really appreciate the contributions of FinRL and would like to point out the uniqueness and advantages of our own TradeMaster at the same time.
> > >
> > > ---
> > > **Q2. For the AI4Finance community, people have really different backgrounds. You should put some effort on how to explain the benefit of RL for Finance to the community.**
> > >
> > > We would like to clarify on this with the following four points:
> > > * While discussing the potential impacts of TradeMaster (line 351-358), we explain how users from different communities (e.g., RL researchers, finance experts, industry financial practitioners and individual investors) can benefit from TradeMaster.
> > > * As far as we know, users from different communities have already involved into the project and contributed to TradeMaster in many ways. For instance, our industry collaborators provide suggestions from the industry production perspective and help us make the simulation environments more realistic. Many undergraduate and Master students have used TradeMaster as the code base of their projects for degree thesis and help us improve the tutorials and software documents. In addition, some researchers start to evaluate their new RL for trading algorithms using the evaluation and visualization toolkits in TradeMaster. Their feedback helps us a lot to improve the quality of the evaluation and visualization components.
> > > * For educational purpose, we provide dozens of hands-on Jupyter notebook tutorials and detailed software documents of TradeMaster. We also give tutorial talks and host symposiums on academic conferences (e.g., AAAI23, KDD23 and IJCAI23) to further benefit more users and enlarge the RL for finance community.
> > > * We include an AutoML component and develop a GUI user interface for users unfamiliar with coding and RL to explore the potential of RL in financial trading.
> > >
> > > In the future, we will continuously update and maintain the project based on the feedback from different communities to improve its user experiences and accessibility. We will definitely further improve this paper based on all discussions if accepted. Thanks again for your time. We are happy to engage in further discussions.
> > >
> > > Best regards,
> > >
> > > Authors of submission 141

---

> ### Author Response · Authors · 2023-08-21
> **Response to Reviewer Vtxm**
>
> **Q3. The MDP steps is really unclear to me. For example, the state and action space of different tasks are obviously different, but the authors only briefly mention it without details. Figure 2 is a general figure, not a specific figure showing how the reward will be facing different actions.**
>
> * To address the confusion on the MDP steps, we include a concrete example to further clarify how it works (line 116-122). Considering a simple trading scenario with only one stock, we obtain $p_t^c$ and $p_{t+1}^c$, which denote the close price of the stock at time $t$ and $t+1$, from historical market data. The action at time $t$ is to buy $k$ shares of the stock. Then, the reward $r_t$ at time $t$ is the account profit defined as $k*(p_{t+1}^c-p_t^c)$. For state, we use historical market data to calculate technical indicators as external state and investors’ private information such as remaining cash and current position is applied as internal state. Note that similar procedures have been widely applied in many existing RLFT work [2, 3].
> * As a holistic trading platform, TradeMaster covers a wide range of trading tasks. It is impossible to provide detailed MDP formulation here due to page limit. We choose to offer a general version MDP formulation (as shown in Figure 2) and point out key uniqueness of different work in our descriptions (line 95-112).
> * For the reward function design, the essence is to calculate the change of capitals. Different work considers various transaction cost (e.g., commission fee [3]) and trading constraints (e.g., leverage [4]) to make the simulation more realistic.
> * For readers unfamiliar with RLFT, Appendix A and this survey [5] offers a detailed introduction on related terminologies and detailed MDP formulation of each trading task.
>
> [3] Wang et al. Commission fee is not enough: A hierarchical reinforced framework for portfolio management. AAAI. 2021.
>
> [4] Sun et al. DeepSclpaer: A risk-aware reinforcement learning framework to capture fleeting intraday trading opportunities. CIKM. 2022.
>
> [5] Sun et al. Reinforcement learning for quantitative trading. ACM Transactions on Intelligent Systems and Technology. 2023.
>
>
> ---
> **Q4. One of the main problems of RL in QT is overhead/time. I would like to see the performance comparison between TradeMaster and other open-sourced work under the same hardware (e.g., RTX 3090).**
>
> We agree that overhead/time is one key challenge of RL in QT especially in the high frequency regime, where the amount of data explodes and the trajectory is extremely long. We describe our efforts on algorithm implementation to improve its efficiency:
> * To provide high-quality implementation for industry-level financial trading applications, we follow the successful development procedures in Stable-Baselines3 [6]. We first carefully read the papers, software documentations and code of other implementations (if available). Then, we implement the algorithms under the efficient components design of TradeMaster with user-friendly interfaces. Moreover, most core functions are covered by unit tests to verify the correctness.
> * For TradeMaster, it takes at most a few hours for most algorithms to converge during training. For inference, it takes only about 10 millisecond to make investment decisions, which shows the potential to deploy on real-time data stream.
> * As for comparison with other trading platforms, it is hard to provide a fair comparison because the different representation of feature, the different lengths of episodes, the design of the loss function and many other details in RL training. Generally speaking, we are confident that TradeMaster is more efficient comparing to existing platforms based on the above discussions.
>
> [6] Raffin et al. Stable-baselines3: Reliable reinforcement learning implementations. Journal of Machine Learning Research. 2021.
>
> ---
> *We would appreciate it if reviewers can confirm that their concerns had been addressed, if so, reconsider their assessment. We’d be happy to engage in further discussions.*

---

> ### Author Response · Authors · 2023-08-24
> **Does our response address your concerns?**
>
> Dear reviewer Vtxm,
>
> Thanks again for your valuable feedback and insightful suggestions. During the rebuttal phase, we carefully go through them, revise the paper and find that there might be some misunderstandings. To address the concerns, we highlight the uniqueness of TradeMaster and point out our efforts and achievements on serving TradeMaster as an open-source project. We also include discussion on the MDP formulation of RL for financial trading with concrete examples and talk about our thoughts on improving the algorithms efficiency.
>
> Could you please take some valuable time to have a look at our responses? We would really appreciate it if you can confirm that the concerns have been addressed and, if so, reconsider the assessment. We are happy to engage in further discussions.
>
> Best regards,
>
> Authors of Submission 141

---

> ### Author Response · Authors · 2023-08-30
> **Urgent request for reviewer's reply on our responses**
>
> Dear reviewer Vtxm,
>
> As it is close to the last day of the rebuttal phase, we would like to request for your reply on our responses. We have spent significant amount of time on carefully going through your feedback, revising the paper and responding to questions. We summarize major points as follows:
> * We highlight our efforts and achievements on serving TradeMaster as an open-source project that benefits different communities.
> * We discuss the uniqueness of TradeMaster with 6 points and include a table to compare with existing work.
> * We provide concrete examples and more details on the MDP formulation of financial trading tasks to improve the clearness of the formulation.
> * We describe our efforts on algorithms implementation to improve the efficiency.
>
> Could you please take some valuable time to have a look at our responses? We are fairly confident that your concerns have been addressed in the new version. It could be a pity for us if our carefully prepared responses are not taken seriously by the reviewer, which hurt the reputation of NeurIPS. We would really appreciate it if you can confirm that the concerns have been addressed and, if so, reconsider the assessment. We are happy to engage in further discussions.
>
> Best regards,
>
> Authors of Submission 141

---

> > ### Comment · Reviewer_Vtxm · 2023-08-31
> > **Respond to the authors**
> >
> > I have carefully reviewed all the rebuttals and I think the authors addressed most of my concerns. I decide to change my previous rating and thank you all for the effort.

---

### Author Response · Authors · 2023-08-21
**General Response and Updates for Revision**

We thank all reviewers for their valuable feedback. Based on their comments, we have revised the paper (orange part) to address concerns and summarize major changes here:
* We include a concrete example to further clarify how the MDP steps work for financial trading tasks (line 116-122). We further explain why a general MDP formulation is provided here and recommend this survey paper [1] for readers interested in detailed formulations for each trading task (line 123-127). [Reviewer Vtxm, DHLR]
* We highlight our efforts and list achievements of TradeMaster as an impactful open-source project, which benefits users from different communities in Appendix C. [Reviewer Vtxm, k4uA]
* We add a paragraph (line 337-349) to point out the uniqueness of TradeMaster and compare it with other popular trading projects in Table 3. [Reviewer Vtxm, fVic]
* More details on baseline selection, experimental setup, hyperparameters and experiments with standard deviation, convergence curve and results analysis are included (line 243 and Appendix F). [Reviewer DHLR, fVic]
* We update contents on TradeMaster GitHub repository, official website and software documents based on reviewers’ feedback. [Reviewer Vtxm, DHLR, k4uA, fVic]

[1] Sun et al. Reinforcement learning for quantitative trading. ACM Transactions on Intelligent Systems and Technology. 2023.

---

### Decision · Program_Chairs · 2023-09-22

**Decision:**

Accept (Poster)

**Comment:**

The authors present an open-source platform that aims to tackle the challenges of reinforcement learning in financial trading (RLFT).

The reviewers generally agreed that the contribution is significant, and that the work will be a valuable tool for RLFT applications.
The reviewers especially noted the strong execution of this work, and the clarity of the paper.

While there were initial concerns, the authors put in significant effort during the author response period, and in the end all reviewers were positive about the paper. I recommend acceptance.